# Learning to Reason With Mixture Of Tokens

**Adit Jain**[1,2] and **Brendan Rappazzo**[1]

[1]Machine Learning Research, Morgan Stanley, New York, NY
[2]School of Electrical and Computer Engineering, Cornell University, Ithaca, NY
aj457@cornell.edu, brendan.rappazzo@morganstanley.com

## Abstract

Reinforcement learning with verifiable rewards (RLVR) has become a leading approach for improving large language model (LLM) reasoning capabilities. Most current methods follow variants of Group Relative Policy Optimization, which samples multiple reasoning completions, scores them relative to each other, and adjusts the policy accordingly. However, these approaches invariably sample discrete tokens at each reasoning step, discarding the rich distributional information in the model's probability distribution over candidate tokens. While preserving and utilizing this distributional information has proven beneficial in non-RL settings, current RLVR methods seem to be unnecessarily constraining the reasoning search space by not using this information. To address this limitation, we investigate mixture-of-token generation (MoT-G) in RLVR. We present a unified framework that generalizes existing MoT-G approaches, including existing training-free methods that construct mixture embeddings as weighted sums over token embeddings, and extend RLVR to operate directly in this continuous mixture space for generating chain-of-thought. Evaluating two MoT-G variants on Reasoning-Gym, a suite of reasoning-intensive language tasks, we find that MoT-G methods achieve substantial improvements (5–35% gains on 7/10 tasks) compared to standard decoding with the Qwen2.5-1.5B model, while reaching comparable accuracy with half the number of trajectories, suggesting improved training efficiency. Through comprehensive hidden-state and token-level analyses, we provide evidence that MoT-G's benefits may stem from its ability to maintain higher hidden-state entropy throughout the reasoning process and promote exploration in token space.

## 1 Introduction

Reinforcement learning with verifiable rewards (RLVR) has emerged as a dominant paradigm for enhancing the reasoning capabilities of large language models. In this approach, models are provided with a "scratchpad" to develop their reasoning step-by-step before producing final answers, with training focused solely on outcome-based rewards while allowing the model to autonomously discover effective reasoning strategies (Shao et al., 2024; Kambhampati et al., 2025). Recent advances in RLVR have demonstrated remarkable success across challenging domains, from achieving gold medal-level performance on International Mathematical Olympiad problems Huang and Yang (2025) to saturating previously difficult benchmarks in logical reasoning Guo et al. (2025). These achievements highlight the potential of allowing models to develop their own reasoning strategies rather than constraining them to human-prescribed approaches, suggesting that RLVR represents a fundamental advancement in how we train LLMs for complex reasoning tasks.

Most current RLVR implementations rely on group relative policy optimization (GRPO) and similar techniques. Which all basically operate by, for a given question, using discrete token sampling to generate some $G$ number of completions or trajectories (reasoning scratchpad + final answer), scoring those final answers, and using the relative scores of each rollout to train the LLM to give better answers (Zheng et al., 2025; Liu et al., 2025b). These methods have proven to work on a variety of reasoning intensive tasks (Stojanovski et al., 2025; Liu et al., 2025a; Guo et al., 2025), and there has been substantial research effort to improve this flavor of RL algorithm, from better normalization methods, to expanding it to multi-turn and tool use trajectories.

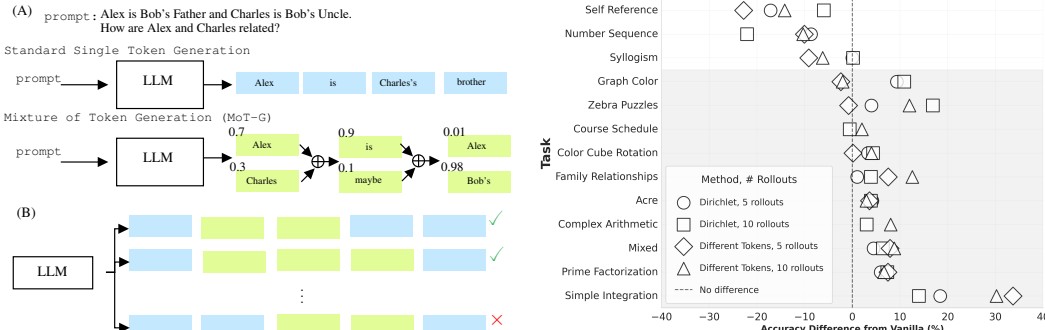

(1) Mixture of Tokens based Generation (MoT-G)   (2) Empirical performance of 2 MoT-G methods

Figure 1: (1.A) We propose a generalized abstraction for Mixture of Token based Generation (MoT-G) which allows autoregressively sampling multiple tokens and aggregating them as a mixture embedding instead of a single token embedding. (1.B) The LLM learns to use mixture of tokens for thinking by performing RLVR finetuning using GRPO. (2) Our experimental results benchmark two variants of MoT-G for finetuning Qwen2.5-1.5B on 10 different reasoning gym tasks with 5 and 10 trajectories. Our methods obtain decent gains $(5-35\%)$ on majority of environments (shaded).

However, much less attention has been paid to a fundamental assumption underlying these approaches: the commitment to discrete tokens at each reasoning step. This discrete sampling paradigm forces models to make early, irreversible decisions in token space, limiting their ability to maintain uncertainty and explore alternative reasoning paths during critical intermediate steps of complex problem-solving. Although some prior work has explored continuous or mixture token representations in language modeling (Zhang et al., 2025b; Deng et al., 2024), there has been limited large-scale studies systematically investigating the integration of mixture-of-token generation within the RLVR training framework (Butt et al., 2025). Furthermore, existing work has not focused on understanding the mechanistic benefits that mixture token representations might provide for reasoning tasks, particularly in terms of exploration capabilities and training efficiency. This represents a significant gap in our understanding of how token-level design choices impact effectiveness of reinforcement learning for reasoning and this paper addresses the following question:

> **Research Question:** *How does using a mixture of tokens, instead of a single token, for generating reasoning steps affect exploration, training efficiency, and downstream performance in reinforcement learning for complex language-based reasoning intensive tasks?*

To study this research question, we introduce mixture-of-token generation (MoT-G) for RLVR, a framework that enables models to maintain distributions over multiple tokens at each reasoning step rather than committing to discrete selections. Our main contributions can be summarized as:

**1. Generalized framework for mixture of tokens based generation (MoT-G).** We introduce MoT-G for RLVR, a unified framework that enables mixture-of-token generation within reinforcement learning paradigms for reasoning, generalizing training-free approaches and more importantly extending RLVR to operate in continuous mixture spaces rather than discrete token selections.

**2. Empirical Results.** We numerically analyze the efficacy of our work on a variety of reasoning tasks from Reasoning-Gym (Stojanovski et al., 2025) in Section 4. We provide comprehensive empirical evaluation demonstrating significant performance improvements across reasoning tasks, with MoT-G achieving 5–35% gains on 7 out of 10 tasks while requiring half the trajectories to reach comparable performance, suggesting improved training efficiency. We present experiments ablation studies on different model sizes, hyperparameters etc. in the Appendix B.

**3. Effect of MoT-G on information in the hidden states and exploration.** We offer mechanistic insights into why MoT-G benefits reasoning through hidden-state and token-level analyses, showing that the approach maintains higher entropy and promotes exploration in token space, providing a deeper understanding of how continuous representations enhance reasoning capabilities.

**Notation.** We denote vectors by bold-face small case letters, $\mathbf{x}, \mathbf{y}..$, matrices and tensors by bold-face capital case letters $\mathbf{X}, \mathbf{Y}..$ and sets by curly letters $\mathcal{X}, \mathcal{Y}...$ Denote $\mathcal{P}(\cdot)$ to be the probability simplex of the set. $\mathbb{P}$ and $\mathbb{E}$ denote the probability and expectation over the appropriate measure.

## 2 BACKGROUND: RELATED WORK AND PRELIMINARIES

**Chain-of-Thought and Scratchpad Reasoning.** It has long been established that large language models demonstrate substantially improved performance when provided with intermediate reasoning steps, commonly referred to as "scratchpads" or chain-of-thought prompting (Wei et al., 2023).

Formally, for an input sequence of embeddings (prompt), $\mathbf{X}_0 = (\mathbf{x}_{-m+1}, \ldots \mathbf{x}_0)^\mathsf{T} \in \mathbb{R}^{d \times m}$, where $m$ is the prompt length, rather than directly predicting a final answer $y$, the models generate an intermediate reasoning trajectory $\tau = (\mathbf{x}_1, \mathbf{x}_2, \ldots, \mathbf{x}_n)$ where each $\mathbf{x}_t$ represents the embedding corresponding to the token of the $t$-th reasoning step. This is followed by the final answer $y$. This can be expressed as: $\mathbb{P}(y|\mathbf{X}_0) = \sum_\tau \mathbb{P}(y|\tau, \mathbf{X}_0)\mathbb{P}(\tau|\mathbf{X}_0)$ where $\mathbb{P}(\tau|\mathbf{X}_0) = \prod_{t=1}^T \mathbb{P}(\mathbf{x}_t|\mathbf{x}_{<t}, \mathbf{X}_0)$ represents the probability of generating the reasoning chain $\tau$. This approach has proven remarkably effective across diverse reasoning tasks, from mathematical problem-solving to logical inference, fundamentally changing how we approach complex reasoning with language models.

**Reinforcement Learning with Verifiable Rewards (RLVR).** GRPO was first successfully demonstrated in DeepSeekMath (Shao et al., 2024) and has since improved reasoning across models including DeepSeek R1, Nemotron, and Phi-4. The key insight is that models can learn to develop their own effective reasoning strategies through outcome-based training alone. Rather than requiring human supervision of intermediate reasoning steps, RLVR enables models to discover optimal reasoning pathways by sampling multiple completion trajectories and learning from final outcomes.

The objective in RLVR is to optimize a policy $\pi_\theta$ (parameterized by $\theta$) to maximize expected return,

$$\pi^* = \arg\max_\theta \mathcal{J}(\theta) = \arg\max_\theta \mathbb{E}_{\tau \sim \pi_\theta} \{r(\tau)\}$$

where $\tau$ represents a complete reasoning trajectory and $r(\tau) : \mathcal{T} \rightarrow \mathbb{R}$ is the reward function defined on the space of trajectories $\tau$. Most current implementations employ Group Relative Policy Optimization (GRPO), which computes advantages relatively within groups of sampled trajectories:

$$\mathcal{J}_{\text{GRPO}} = \mathbb{E}_{(\tau_1, \tau_2, \ldots, \tau_G) \sim \pi_{\theta_{\text{old}}}} \left\{ \frac{1}{G} \sum_{g=1}^G \frac{1}{|\tau_g|} \sum_{t=1}^{|\tau_g|} A_{g,t} - \beta \mathbb{D}_{\text{KL}}[\pi_\theta || \pi_{\text{ref}}] \right\}$$

where the approximate advantage for trajectory $g$ at step $t$ is computed as $A_{g,t} = \frac{r_g - \mu}{\sigma}$, with $r_g$ being the reward of trajectory $g$, $\mu$ the mean group reward, and $\sigma$ the standard deviation.

This approach has catalyzed the current wave of multi-turn reasoning systems, tool-use capabilities, and agentic AI applications by demonstrating that sophisticated reasoning behaviors can emerge from simple outcome optimization. Although there has been research critically analyzing RLVR's complete utilization of base model potential (Yue et al., 2025) and reasoning consistency (Shojaee et al., 2025), recent empirical (Liu et al., 2025a) and theoretical (Wen et al., 2025) results demonstrate that GRPO incentivizes correct reasoning paths and logical integrity.

**Continuous and Mixture of Token based Generation.** Parallel to advances in reinforcement learning for reasoning, a separate line of research has questioned the fundamental assumption of discrete token sampling in autoregressive generation. COCONUT (Hao et al., 2024) uses distillation to replace chain-of-thought with continuous embeddings, with theoretical work showing polynomial-time solutions through token superposition (Zhu et al., 2025a). Training-free methods include soft-thinking (Zhang et al., 2025b), which averages token embeddings weighted by probabilities, and approaches using VQ-VAE compression (Su et al., 2025), speculative soft tokens (Xu et al., 2025), embedding optimization (Zhu et al., 2025b), and Bayesian posterior mixing (Zhuang et al., 2025). The Coconut paper (Hao et al., 2024) and related mixture-of-tokens work (Zhang et al., 2025b) have demonstrated that such approaches can improve generation quality and enable more nuanced reasoning in standard (non-RL) settings. However, none have systematically studied mixture generation within reinforcement learning frameworks for reasoning tasks.

At each generation step, language models produce a rich probability distribution $\mathbb{P}(\cdot|z_{<t})$ over the vocabulary $\mathcal{D}$, yet standard generation discards this distributional information by sampling a single token $z_t \sim \mathbb{P}(\cdot|z_{<t})$ and feeding only its one-hot representation forward. Recent work has explored preserving this distributional information through mixture-of-token approaches. In soft-thinking (Zhang et al., 2025b), mixture embeddings are constructed as weighted combinations over all vocabulary tokens: $\mathbf{x}_{\text{mix}} = \sum_{z \in \mathcal{D}} \mathbb{P}(z|\mathbf{x}_{<t})\mathbf{x}^z$ where $\mathbf{x}^z$ represents the embedding of token $z$. More generally, mixture-of-token generation (MoT-G) methods maintain representations that preserve uncertainty and alternative paths rather than committing to discrete selections. Formally, instead of standard discrete generation, MoT-G uses continuous representations: $\mathbf{x}_{\text{mix}} = \texttt{aggregate}(\texttt{sample}(\mathbf{x}_{\text{mix},<t}))$, where $\texttt{sample}$ and $\texttt{aggregate}$ are defined later.

**Combining RLVR with Mixture Generation.** Despite the demonstrated benefits of both RLVR for reasoning and mixture-of-token approaches for generation quality, no prior work has systematically investigated their combination. This represents a significant missed opportunity: while RLVR methods excel at discovering effective reasoning strategies through outcome optimization, they may be unnecessarily constraining the exploration space by forcing discrete commitments at each reasoning step. Conversely, mixture approaches have shown promise for preserving reasoning flexibility but have not been studied in the context of reinforcement learning where models must learn to balance exploration and exploitation across extended reasoning trajectories. The central question we address is whether extending RLVR to operate in continuous mixture spaces can enhance both reasoning performance and training efficiency, potentially allowing models to defer commitments and maintain consideration of alternative reasoning paths throughout the learning process.

## 3 MIXTURE OF TOKENS GENERATION (MOT-G)

We now describe one of our key contribution, a generalized framework for mixture of token generation. Put simply, we replace the embedding of the sampled token with a mixture of $k$ tokens. Consider a large language model $\mathcal{L}$ which has vocabulary $\mathcal{D}$ and context length $\mathsf{N}$. Let (input) token embeddings be mapped by the function $l : \mathcal{D} \to \mathcal{X}$ and let $\mathbf{X}_t = (\mathbf{x}_1, \ldots, \mathbf{x}_t) \in \mathcal{X}^t$ denote a tensor of token embeddings corresponding to a sequence of length $t$. Denote the transformer function of $\mathcal{L}$ by $h : \mathcal{X}^{\mathsf{N}} \to \mathcal{P}(\mathcal{D})$ which takes in input as token embedding tensor and outputs a probability distribution over the vocabulary and denote $\mathbf{p}_t \in \mathcal{P}(\mathcal{D})$ to be the output at generation step $t$.

Recently a technique titled "soft-thinking" was introduced which uses a weighted average of token embeddings during the generation phase (Zhang et al., 2025b). We provide recipe for generating using a mixture of token embeddings which generalizes soft-thinking and allows for a wide-variety of choices to design a generation approach. Note that any standard generation technique can be derived as a special case of this framework. There are four key steps in the generation process - $\texttt{initialize}$ - for initialization, $\texttt{sample}$ - sampling, $\texttt{aggregate}$ - for aggregation and $\texttt{endcriteria}$ - for end criteria. We now describe the mixture of token generation:

**Mixture of Token Generation (MoT-G):** $\texttt{initialize}$ $\mathbf{X}_0 = (\mathbf{x}_{-m+1}, \ldots \mathbf{x}_0)^{\mathsf{T}} \in \mathbb{R}^{d \times m}$. Do the following for $t = 1, 2, \ldots$ until $\texttt{endcriteria}(\mathbf{X}_t)$.

**Step 1.** Sample $k$ tokens $\mathcal{S}_t = \texttt{sample}(\mathbf{p}_t = h(\mathbf{X}_{t-1}))$.

**Step 2.** Get embedding $\mathbf{x}_t = \texttt{aggregate}(\mathcal{S}_t, h(\mathbf{X}_{t-1}))$. Append $\mathbf{x}_t$ to end of $\mathbf{X}_t$.

After the loop terminates, do standard generation (described above) till the end of sequence token.

We now discuss $\texttt{initialize}, \texttt{endcriteria}, \texttt{sample}$ and $\texttt{aggregate}$ in detail.

*Possible Initialization and End Criteria:* One approach is to start answering directly using the chat template (if available) that the model was instruction finetuned on. Similar to soft-thinking (Zhang et al., 2025b), one can add a think token (e.g. $\texttt{<think>}$) at the beginning of the generation. Further akin to the think step by step prompt (Wei et al., 2023), one can add prompts which encourage the model to consider contrasting options (e.g. think about different approaches to solve).

The end-criteria decides when the mixture of token generation ends and standard generation resumes. One way is to generate till a end think token (e.g. $\texttt{</think>}$ token) is the most-likely token. Another technique which the soft-thinking takes is to stop when the entropy of $\mathbf{p}_t = h(\mathbf{X}_t)$ is below a certain threshold (chosen hyperparameter) for consecutive rounds (another hyperparameter) and

perform parameter sweeps for the optimal value of the entropy threshold (Zhang et al., 2025b). Note that setting the hyperparameters is a challenging task dependent on the problem instance, which can be subsided if using RL which allows the LLM to learn when to stop using MoT-G.

*Sample Function:* The sample function $\texttt{sample} : \mathcal{P}(\mathcal{D}) \to 2^{\mathcal{D}}$ takes in as input the probability distribution and returns $k$ tokens. Previous research in the training-free regime (Zhang et al., 2025b; Zhuang et al., 2025) consider all tokens and change the aggregation mechanism (described next), however for ensuring diversity across trajectories we consider a subset of tokens as the input. There are different sampling techniques one can come up with and we discuss a few.

1. *Top-k sampling* This is extension of greedy sampling to the mixture domain, where the tokens with the $k$ highest probabilities are selected.

2. *min-p sampling*: Sample $k$ tokens without replacement from the subset of tokens which have probabilities greater than $p$ (hyperparameter), with normalized probabilities.

3. *Nucleus sampling*: Sample $k$ tokens from the subset of tokens with cumulative probability greater than some threshold (hyperparameter).

4. *SWR $k$ sampling*: Sample $k$ tokens without replacement (SWR) with some temperature (similar to normal sampling in standard autoregressive generation).

*Aggregate Function:* The aggregate function, $\texttt{aggregate} : 2^{\mathcal{D}} \times \mathcal{P}(\mathcal{D}) \to \mathbb{R}^d$ aggregates embeddings of the sampled vectors into a single mixture embedding. There are many different approaches that one can take to aggregate $k$ tokens into a single token embedding. A more general framework would map $k$ tokens into $m \ (< k)$ tokens, and future work can look at such an approach bridging the gap between this paper and COCONUT where embeddings are learned to compress think tokens across generation steps. We now highlight a few obvious choices for aggregation.

The most obvious choice is to consider a weighted sum of the sampled tokens with different weighing choices, $\sum_{i \in |\mathcal{S}_t|} w_i \mathcal{S}_t[i]$, where $w_i$ are the weights for $i$-th token in the sampled set $\mathcal{S}_t$. One can weigh the embedding uniformly by $1/k$, effectively taking the bary-centre of the sampled tokens. Further one can weigh the embeddings by *normalized probabilities* (Zhang et al., 2025b), which allows more representation of tokens with higher probabilities. Zhuang et al. (2025) comes up with a more sophisticated aggregation mechanism where the Bayesian posterior on the weighted sum to be used is updated based on the sampled token and the probability distribution. One could also weigh the vectors by randomly sampled weights from a suitable distribution, e.g. *Dirichlet distribution* with the normalized probabilities as the support parameter. Such an approach allows for better representation of more likely tokens on average but with added randomness.

Although we consider weighted aggregation, there are other options for aggregation, for e.g., element-wise maximum of the sampled embeddings, which lies on the convex hull. We benchmark aggregation and sampling schemes on GSM-8K in the training-free regime in Appendix B.

## 3.1 Adapting GRPO for MoT-G

We now discuss how GRPO can be modified for generating reasoning tokens with MoT-G.

*Formatting and Rewards.* Similar to standard generation, we consider using specific tokens for letting the model produce its chain of thought and once the end think token is the most-likely token to stop the mixture generation. The answer of the model is extracted from a specified format. Since the answer is generated after the mixture based generation ends, one can straightforwardly give exact-match or LLM-as-a-judge correctness rewards. Similarly the length of reasoning can be rewarded. However, any other reward which is based on rewarding the content of the reasoning chain (for e.g. using a LLM judge) and is sensitive to coherence might need some additional engineering difficult to reward in practice, which is a limitation of this method. One possible workaround is sampling a token from the set of tokens being mixed and reward this auxillary chain of thought.

*Loss Computation.* However the computing loss for such a generation technique is more subtle since at each step there are multiple tokens over which the loss can be propagated. We disambiguate between two different ways one can compute loss for the mixture based generation, each of them is significant in our opinion and we study a variant of each of them experimentally.

*Single token losses:* One can propagate the loss using the log probabilities of a single sampled token from the set of tokens used for generating mixture embeddings, $z_t \sim \mathcal{S}_t$. The log probabilities of this token can either be left unweighted or weighed by their respective probabilities.

*Multiple token losses:* One can propagate the loss is to each of the tokens in $\mathcal{S}_t$. For on-policy RL without stabilization ($\pi_{\text{old}} = \pi_\theta$), the multiplicative factor does not matter for the advantage term, as long as the appropriate tokens are gathered. For the KL regularization loss it does create a difference theoretically, however empirically we don't observe one can use a single token approximation.

For on-policy RL with stabilization ($\pi_{\text{old}}$ in Shao et al. (2024)) one needs a plug-in estimate for $\log \pi_\theta(\tau)$. In standard autogeneration when the trajectory is considered the sequence of tokens ($\tau = (z_1, \ldots, z_t)$), one can substitute this as $\log \pi_\theta(\tau) = \log(\Pi_{n=1}^t \mathbb{P}(z_n | (z_1, \ldots, z_{n-1}))) = \log(\Pi_{n=1}^t \mathbf{p}_n(z_n)) = \sum \log \mathbf{p}_n(z_n)$, where we denote $\mathbf{p}_n = \mathbb{P}(z_n | (z_1, \ldots, z_{n-1})))$. Since now the sequence of tokens is replaced by a sequence of mixture of tokens given by the matrix with entries $(z_{m,n})_{k \times t}$. For each generation step $t$, one ideally needs to compute the probability $\mathbb{P}(\{z_{1,t}, \ldots, z_{1,t}\} | \mathbf{x}_1, \ldots, \mathbf{x}_{t-1})$ which can be combinatorially expensive to compute (especially if sampling without replacement). One way to approximate is to sum the log probability of each token weighted by the probability, $\log(\pi_\theta(\tau)) \approx \sum_{m,n} \mathbf{p}_n(z_{m,n}) \log(\mathbf{p}_n(z_{m,n}))$.

## 3.2 Exploration in GRPO with Mixture of Tokens

Directly applying soft thinking (Zhang et al., 2025b) to GRPO based RL in LLMs is challenging because soft thinking does not allow for any randomness, since the probability distribution $\mathbf{p}_t$ over the dictionary and the embedding map $l$ are deterministic (fixed) given the same sequence of embedding vectors, $\mathbf{X}_t$. This leads to zero exploration when doing multiple trajectories in a group. We now describe and motivate two generation techniques which we analyze in our experiments.

**A. Dirichlet stochastic weighing with top-$k$ tokens (Dirichlet).** We sample the top$-k$ tokens greedily and the weights are sampled from a Dirichlet distribution (described in Appendix C.2) with the normalized probability distribution as the parameter. There are two primary reasons for considering this, (a) such a prior allows for exploration and increases the entropy of the mixture input embedding (b) it does so in a controlled fashion, i.e., because the Dirichlet distribution has higher entropy around the centroid of the probability simplex and low entropy near the boundary, the resulting mixture weights are aligned with the model's confidence and do not alter them considerably (see Figure 6 in Appendix). Note that as $k$ increases the weights are more likely to be attributed to unhelpful tokens.

**B. Sampling $k$ tokens without replacement and weighing with normalized probability (Different Tokens).** We now state the following result, which illustrates the fundamental trade-off involved in the choice of $k$ for this MoT-G generation method,

**Proposition 1.** *Consider the LLM $\mathcal{L}$ described above with dictionary $\mathcal{D}$, token embedding space $\mathcal{X}$. We consider mixture of token generation using sampling $k$ tokens without replacement and aggregating them using a weighted sum with their normalized probability. Let $G$ be the number of trajectories. Let $L_t$ be the number of unique tokens across the trajectories at generation step $t$ and $\mathbf{x}_g \in \mathcal{X}$ to be the generated mixture embedding for trajectories $g \in [G]$. With the expectation taken over the generation process, $\mathbb{E}\{L_t\}$ is a monotonically increasing function in $k$. Further, $\mathbb{E}_{g,g' \sim [G]}\left\{\|\mathbf{x}_g - \mathbf{x}_{g'}\|^2\right\}$ is a monotonically decreasing function in $k$.*

The proof is in Appendix C and follows straightforwardly using arguments from probability theory. This shows the tradeoff inherent to sampling $k$ tokens - we improve the coverage of tokens across mixtures as $k$ increases but increases the overlap in the resulting embedding mixture. Therefore it is suitable to choose a reasonably small value of $k$, and we do our experiments primarily with $k = 2$.

## 4 Numerical Experiments with Reasoning-Gym

We demonstrate how our method using the two generation methods described in the last section perform better than Single-Token GRPO in a variety of different environments. Out of the 10 environments, our methods improve performance in 5 environments and perform at par with just 5 chains when compared to standard generation with 10 chains, hence significantly improving diversity and allowing better scaling. We fix the model to be Qwen2.5 with 1.5 billion parameters to understand

how our method compares to Single-Token GRPO in an isolated setting. We conclude the section with analysis of the hidden states using recent methods in interpretability to infer why our methods allow for better exploration. We provide supplementary experiments and ablations in Appendix B.

To analyze the performance of model's reasoning performance when trained with MoT-G, we need a evaluation harness which contains examples which are not in-distribution of the training data so that there is enough room for improvement and one can illustrate improvement in sampling correct reasoning paths. We benchmark *Dirichlet* standard math datasets including Math-500 and GSM-8K in Appendix B where our methods perform competitively but gains are minimal $(1-2\%)$. However we primarily demonstrate the efficacy of our method on a recent released collection of procedurally generated reasoning tasks, titled, *Reasoning-Gym* (Stojanovski et al., 2025). Our main results are benchmarked on 10 tasks across 5 different categories, prime factorization (math), simple integration (math), family relationships (graphs), acre (induction), color cube rotation (cognition), graph color (algorithmic), syllogism (logic), self reference (logic) and number sequence (logic) and mixed (color cube rotation, prime factorization, shortest path, acre, graph color, family relationships).

*Experiment Methodology* We first fixed the primary experimental setup: We used the instruction finetuned version of Qwen2.5 with 1.5 billion parameters. Appendix B has experiments on other model sizes of the same family. The number of learning steps was fixed to 1000 steps, the learning rate to $10^{-6}$, the KL regularization parameter to 0.2, the batch size was 1, the maximum generation steps and prompt length were both fixed to 1024 for all experiments. We evaluate on a holdout set every 50 iterations with 100 samples. We selected these 10 tasks because the other tasks we tried on (specifically maze, sudoku, shortest path, dice, course schedule) do not show learning with the fixed parameters (with our method *or* standard generation). We benchmark 3 methods: (a) standard generation and standard GRPO loss (*single-token*) (b) Sampling top-$k$ tokens, aggregating them with weights sampled from Dirichlet distribution with the normalized probability as parameter and loss on single token (*Dirichlet*) (c) Sampling $k$ tokens and aggregating tokens with normalized probability weights and propagating loss to all the sampled tokens (*Different Tokens*).

## 4.1 Aggregate Analysis on Reasoning Gym Tasks with Qwen2.5-1.5B

| Task | 5 Chains | | | 10 Chains | | |
|------|------|------|--------|------|------|--------|
| | Dir | DT | Single | Dir | DT | Single |
| Acre | **42.7 ± 3.4** | 42.0 ± 2.4 | 38.5 ± 0.2 | **42.5 ± 0.5** | 41.5 ± 9.1 | 38.6 ± 1.2 |
| Color Cube Rotation | **24.5 ± 1.0** | 21.3 ± 1.9 | 21.2 ± 0.2 | **25.5 ± 0.8** | 25.2 ± 1.5 | 21.2 ± 0.8 |
| Graph Color | **33.1 ± 1.3** | 21.4 ± 12.1 | 23.8 ± 1.1 | **36.1 ± 1.5** | 23.2 ± 14.2 | 25.2 ± 1.3 |
| Mixed | 29.3 ± 1.6 | **32.8 ± 1.6** | 24.9 ± 2.5 | 31.5 ± 2.3 | **33.9 ± 1.9** | 25.1 ± 0.9 |
| Family Relationships | 40.3 ± 4.3 | **46.8 ± 3.6** | 39.3 ± 1.5 | 47.0 ± 0.6 | **55.7 ± 8.3** | 43.2 ± 2.9 |
| Simple Integration | 44.0 ± 2.6 | **59.3 ± 1.9** | 25.6 ± 2.1 | 57.5 ± 3.4 | **73.8 ± 3.2** | 43.6 ± 4.9 |
| Prime Factorization | 78.2 ± 0.6 | **79.7 ± 4.4** | 72.3 ± 1.5 | 83.6 ± 3.9 | 82.9 ± 2.9 | 76.3 ± 0.5 |
| Syllogism | **82.5 ± 5.0** | 73.3 ± 12.3 | **82.5 ± 4.5** | 84.6 ± 3.1 | 78.2 ± 13.1 | 84.4 ± 4.1 |
| Number Sequence | 46.3 ± 13.2 | 44.9 ± 4.3 | **54.9 ± 3.4** | 38.9 ± 8.6 | 50.7 ± 7.8 | **61.0 ± 5.9** |
| Self Reference | 13.6 ± 3.3 | 7.9 ± 4.1 | **30.7 ± 0.8** | 24.5 ± 7.6 | 16.2 ± 13.2 | **30.4 ± 0.8** |

Table 1: Accuracy (%) ± standard deviation by task, number of chains, and method. Dir=Dirichlet, DT=Different Tokens, Single=single-token. All values are the reported pass@1, for further significance testing we ran each experiment three times, with three different random seeds, and report the mean and standard deviation.

We summarize the results in Table 1. We present the pass@1 accuracies of the best checkpoint for the different tasks for the different methods with different number of trajectories (groups or chains).

*Insight 1.* We observe that there are three different types of performance behaviors exhibited by our methods. Type A (First 6 tasks): Tasks where both the methods perform significantly better with gains of around $5-40\%$ in the pass@1 accuracy. Further on these tasks the performance using mixture methods perform competitively or better with half the number of chains with modest $(2-10\%)$ gains. This is a desired consequence since it allows for more exploration with the same number of chains which allows for improvement in policy. Type B (Syllogism) : Tasks where one of the methods perform competitively. Type C (Number Sequence and Self Reference): Tasks

where both the mixture underperform the single-token model by roughly $5-20\%$. We speculate that Type C tasks may require more precise, deterministic reasoning steps where maintaining uncertainty through mixture representations introduces harmful noise rather than beneficial exploration.

## 4.2 HIDDEN STATE AND TOKEN-LEVEL ANALYSIS OF THE MOT-G METHODS

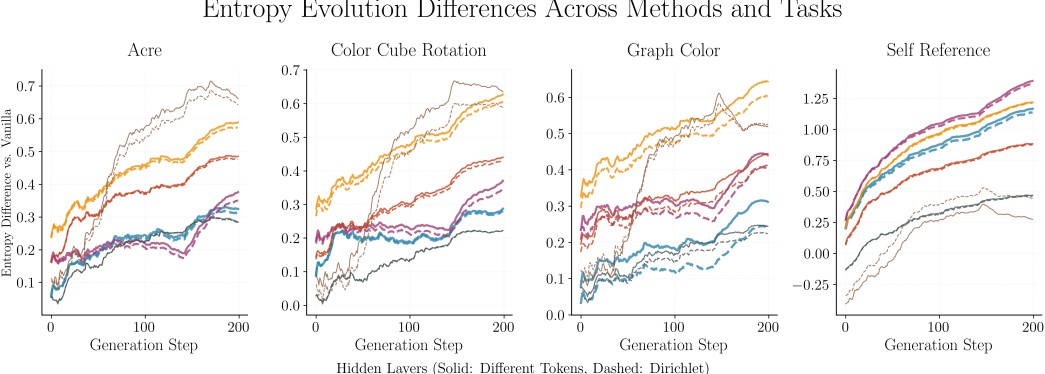

Figure 2: Difference in entropy evolution with respect to single-token over generation steps for different environment. The entropy is computed using the eigenvalue distribution of the Gram matrix of the hidden state vectors of a particular layer. The entropy is representative of the informativeness and diversity of features (Skean et al., 2025). RLVR in LLMs with MoT-G generation generally leads to a higher entropy across layers and generation steps than with single-token generation.

*Analysis of Hidden States.* We use techniques from Skean et al. (2025) to analyze the entropy of eigenvalue distribution of the Gram matrix of the sequence of hidden layer vectors for different generation steps of a particular layer. The entropy has shown to be indicative of the information representation quality, higher entropy indicates more information and more diverse features Skean et al. (2025). Consider the hidden layer $l$ of the LLM. For generation step $t$ let $\mathbf{z}_t^{(l)} \in \mathbb{R}^{d_h}$ be the hidden state vector. Then for the generation steps $t = 1, \ldots, n$, denote the sequence of hidden state vectors $\mathbf{Z}_n = [\mathbf{z}_1^{(l)}, \ldots, \mathbf{z}_n^{(l)}] \in \mathbb{R}^{n \times d_h}$. We consider the eigenvalue distribution of the Gram matrix, $\mathbf{K}_n = \mathbf{Z}_n \mathbf{Z}_n^{\mathsf{T}} \in \mathbb{R}^{n \times n}$. Specifically we consider the von Neumann entropy of the eigenvalue distribution, $S(\mathbf{Z}_n) = -\sum_i \frac{\lambda_i}{\sum_j \lambda_j} \log\left(\frac{\lambda_i}{\sum_j \lambda_j}\right)$, where $\lambda_i$ is the $i$-th largest eigenvalue of $\mathbf{Z}_n$.

We plot the difference in the entropy $S(\mathbf{Z}_n)$ between our mixture methods (Dirichlet and Different-Tokens) and single-token generation in Figure 2 for different total generation steps $n = 1, \ldots, 200$, for 4 different environments for the $l = 1, \ldots, 12$ hiddent layers for Qwen2.5-1.5B model with the the checkpoint of training step 800 with 10 trajectories.

*Insight 2:* The MoT-G Dirichlet and different tokens methods generally have a higher entropy than the single-token generation. Further the gap increases with increasing generation step (tokens) and is present across different layers. The entropy usually plateaus or decreases with generation steps when the latter tokens become less informative. This indicates that MoT-G is able to retain more information in the hidden-layer embedding space than single-token generation, and gives one possible explanation of the improved downstream performance observed when using MoT-G with RLVR.

*Token Diversity Analysis.* We further analyze the number of unique tokens sampled for each training step across the trajectories averaged across the generation steps for different methods in Figure 3.

*Insight 4.* MoT-G methods explore (consider) more unique tokens on average than single-token generation. The Different tokens method has the highest exploration but also has higher variance. Note that both the token count and hidden state entropy is unusually high for the self-reference task (where MoT-G methods perform worst than standard generation; example in Appendix) showing that the model using MoT-G can potentially get confused in more esoteric tallying intensive tasks.

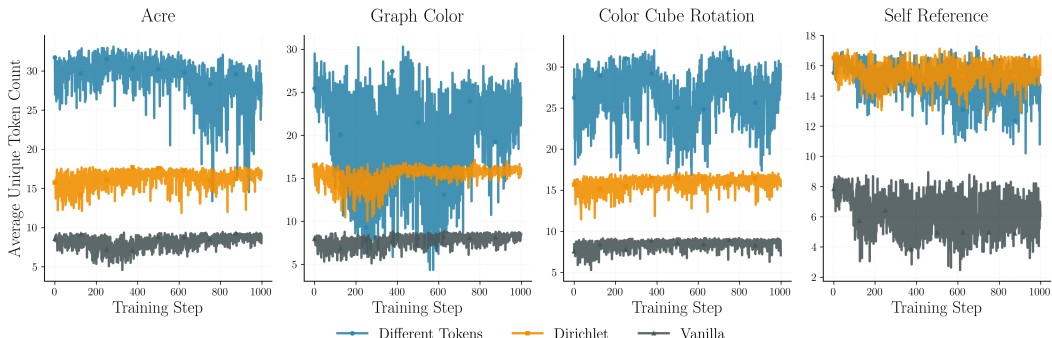

Figure 3: Average unique token counts used, by each method, to generate the next token embedding across trajectories. The unique tokens are indicative of the exploration in the token space.

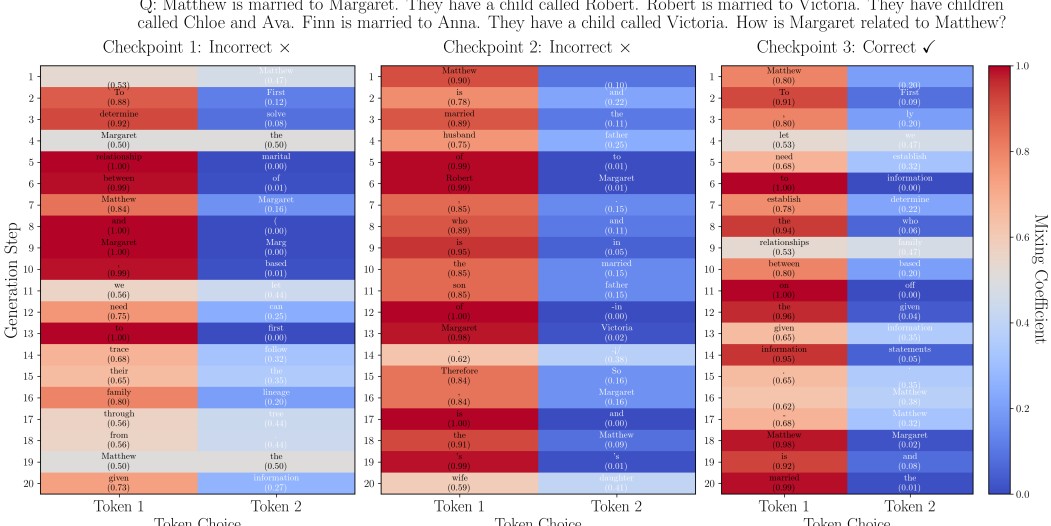

Figure 4: Mixture token weights for a family relationship reasoning problem across training checkpoints. Each row shows a generation step, with the left column displaying the highest-weighted token and its probability, and the right column showing the second-highest weighted token. The model learns to simultaneously consider multiple plausible reasoning paths, such as weighing both "father" and "son" as potential relationships, rather than committing prematurely to a single discrete choice.

*Improvement in Reasoning with Mixture of Tokens.* Figure 4 shows the sampled tokens and corresponding coefficients of mixing for a specific problem of the dataset family-relationships.

*Insight 5.* RLVR with MoT-G enables the LLM to learn how to accordingly weigh different options (e.g. son and father in Step 10 of middle plot in Figure 4) and also keep a consistent belief of the different possibilities (e.g. Victoria v/s Robert, Margaret v/s Matthew, child and son).

## 5   CONCLUSION

This paper investigates mixture-of-token generation as an alternative to discrete sampling in reinforcement learning for reasoning. We present a unified framework for incorporating mixture representations into RLVR and demonstrate the necessary modifications to standard training pipelines. Our evaluation on Reasoning-Gym reveals that MoT-G methods achieve substantial improvements on most tasks (5-35% gains on 7/10 tasks) while requiring half the trajectories to reach comparable performance, though modest regressions occur on some tasks. Analysis suggests these improvements stem from enhanced hidden state informativeness and increased token level exploration during

reasoning. Two promising directions for future work emerge: (a) Curriculum Learning: Investigating whether curricula that progressively increase both task difficulty and mixture size K can yield sustained performance gains across increasingly challenging reasoning problems, and (b) Transfer Learning: Understanding how mixture token representations learned on specific reasoning tasks generalize to new domains and if cross-task knowledge transfer can be improved through MoT-G. The primary limitations are that mixture-of-thought decoding makes it hard to reward the quality of intermediate reasoning and requires approximate, expensive likelihood/KL calculations during training. The added uncertainty can sometimes hurt performance on precise, deterministic tasks.

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

CONTENTS

## A   CONCURRENT SUBMISSION

Concurrent to preparing our arXiv submission, (Butt et al., 2025) (posted September 23, 2025 on arXiv) study 'soft' token chains of thought optimized with RL, introducing noise-injected soft/fuzzy tokens and deriving a REINFORCE training objective, they report gains on math and reasoning benchmarks with Llama/Qwen up to 8B. Our work instead formulates MoT-G inside RLVR, operating on explicit k-token mixtures (e.g., top-k + Dirichlet weights) and training with GRPO on Reasoning-Gym. We also provide mechanistic analyses of hidden state entropy/exploration and

show efficiency gains in trajectories. Given the timing, we provide a brief comparison below in the spirit of research integrity and transparency. We will try our best to update the arXiv submission to numerically compare a version of their approach within our setup.

1. **Mixture object.** *Soft Tokens, Hard Truths:* uses continuous "soft" tokens formed as probability-weighted mixtures in embedding space (plus "fuzzy" variants). *This paper:* proposes a generalized *mixture-of-token* generation mechanism over a discrete subset of $k$ tokens (e.g., top-$k$/nucleus). Their approach can be seen as a special case of our approach with weighted aggregation over the entire dictionary ($k = |\mathcal{D}|$) and added Gaussian noise. We study two completely different MoT-G instances.

2. **RL objective and training loop.** *Soft Tokens, Hard Truths:* derives and trains with a REINFORCE/RLOO-style objective on trajectories containing soft/fuzzy tokens. Their loss function is different and based on the embeddings. *This paper:* instantiates mixture-of-token generation *inside RLVR* using a GRPO/RLVR-style objective tailored to that framework.

3. **Decoding at inference.** *Soft Tokens, Hard Truths:* trains with soft/fuzzy tokens, but reports best results with *hard* token decoding at test time. *This paper:* uses standard hard-token decoding; mixtures are primarily a *training-time* device to enrich exploration within RLVR.

4. **Evaluation focus and metrics.** *Soft Tokens, Hard Truths:* evaluates on math-reasoning benchmarks (e.g., GSM8K, MATH, OlympiadBench) with pass@$k$ metrics on open models. *This paper:* evaluates within the RLVR setting (e.g., Reasoning-Gym) and emphasizes *trajectory efficiency* (accuracy vs. number of trajectories) and analysis of entropy/hidden-state dynamics.

5. **Goal and contribution style.** *Soft Tokens, Hard Truths:* emphasizes feasibility and gains of RL with soft/fuzzy tokens plus guidance on when hard decoding helps. *This paper:* is more expository: it formalizes mixture-of-token generation inside RLVR and provides mechanistic (hidden-state) analyses and efficiency trade-offs.

6. **Hyperparameters:** *Soft Tokens, Hard Truths:* full-probability mixtures with temperature/noise schedules in embedding space. *This paper:* subset size $k$, candidate selection rule (top-$k$ or nucleus), Dirichlet concentration for mixture weights, and GRPO/RLVR hyperparameters.

## B  SUPPLEMENTARY EXPERIMENTS

*Additional Experiment Details:* Most of the experiments were done on a single H100 GPU with 80GB of VRAM, however some experiments were done on multiple such GPUs. We attach slurm scripts and code to run our experiments efficiently.

In general we run all our experiments with default parameters of reasoning gym. Note that for the 3B and 7B models, task we increased the difficulty of color cube rotation (min rotations = 8, max rotations 20), graph coloring (num colors=4,min num vertices=10,max num vertices=20,edge probability = 0.4), family relationship (min family size=10,max family size=20) because the tasks were too easy otherwise.

**Table 2: Qwen2.5-3B.** The MoT-G variants consistently outperform single-token decoding on most tasks, with the Different-Tokens method showing the largest gains on *Family Relationships* ($57.60 \pm 3.11$ vs. $41.27 \pm 2.10$) and solid lifts on *Acre* ($58.40 \pm 2.09$ vs. $51.80 \pm 0.20$). Improvements are modest but positive on *Mixed* and *Graph Color*, while *Color Cube Rotation* is roughly on par across methods. Overall, these results indicate that preserving token uncertainty during the "think" phase benefits relational and inductive reasoning more than low-signal, visually grounded puzzles, aligning with the paper's argument that MoT-G enhances exploration without sacrificing coherence.

**Table 3: Qwen2.5-7B.** Scaling the base model preserves (and often amplifies) MoT-G's advantages: Different-Tokens again leads on *Family Relationships* ($79.30 \pm 2.12$ vs. $73.27 \pm 3.32$) and improves *Acre* ($67.40 \pm 1.83$ vs. $59.87 \pm 3.25$), with both MoT-G variants boosting the *Mixed* suite as well. Absolute accuracies remain low on *Graph Color* for all methods, suggesting algorithmic bottlenecks rather than decoding alone. These results show that mixture-based thinking complements model scale, translating uncertainty-aware hidden states into higher pass@1 without additional trajectories.

**Table 4: Trajectory count (15 vs. 20 chains).** With fewer chains, MoT-G matches or exceeds Single-Token, highlighting sample-efficiency gains. For *Acre*, Different-Tokens improves the 15-chain maximum ($47.1\pm3.2$ vs. $42.1\pm3.7$) and remains ahead at 20 chains ($46.0\pm3.7$ vs. $38.2\pm1.6$). *Prime Factorization* shows strong Dirichlet performance at 15 chains ($86.5 \pm 0.5$ vs. $81.7 \pm 1.4$) and sustained advantage at 20 chains ($85.3 \pm 0.2$ vs. $81.5 \pm 3.0$). In *Number Sequence*, MoT-G is competitive but not uniformly superior, underscoring task dependence. Collectively, the table supports the claim that MoT-G can reach comparable or better accuracy with fewer trajectory by encouraging broader exploration per step.

**Table 5, Table 6, Table 7: Temperature sensitivity.** When decoding temperature rises, Dirichlet-weighted MoT-G remains notably robust while Different-Tokens and Single-Token degrade sharply. At $T = 1.6$ (10 chains) on *Family Relationships*, Dirichlet attains 36.9 vs. 7.1 (Different-Tokens) and 6.2 (Single-Token); at $T = 2.0$ (10 chains) on *Acre*, Dirichlet still reaches 32.7 vs. 0.2 and 8.9, respectively. Similar patterns hold on *Color Cube Rotation*. Because Dirichlet perturbs weights over a stable top-$k$ set (Figure 6), it preserves structure under higher stochasticity, whereas resampling tokens or committing to single tokens collapses.

| Task | Dirichlet | Different Tokens | Single Token |
|---|---|---|---|
| Acre | $55.00 \pm 2.27$ | $\mathbf{58.40 \pm 2.09}$ | $51.80 \pm 0.20$ |
| Color Cube Rotation | $\mathbf{26.30 \pm 5.80}$ | $21.80 \pm 5.37$ | $22.40 \pm 5.39$ |
| Family Relationships | $44.47 \pm 1.63$ | $\mathbf{57.60 \pm 3.11}$ | $41.27 \pm 2.10$ |
| Graph Color | $2.53 \pm 0.12$ | $\mathbf{3.20 \pm 0.72}$ | $1.70 \pm 0.42$ |
| Mixed | $\mathbf{41.89 \pm 3.91}$ | $40.61 \pm 2.67$ | $39.44 \pm 2.30$ |

Table 2: Accuracy (%) with Qwen2.5-3B Dir=Dirichlet, DT=Different Tokens, Single=Single-Token.

| Task | Dirichlet | Different Tokens | Single Token |
|---|---|---|---|
| Acre | $63.33 \pm 1.40$ | $\mathbf{67.40 \pm 1.83}$ | $59.87 \pm 3.25$ |
| Color Cube Rotation | $22.60 \pm 7.81$ | $20.65 \pm 8.17$ | $\mathbf{23.27 \pm 7.09}$ |
| Family Relationships | $72.40 \pm 2.11$ | $\mathbf{79.30 \pm 2.12}$ | $73.27 \pm 3.32$ |
| Graph Color | $2.93 \pm 0.81$ | $\mathbf{4.00 \pm 1.60}$ | $2.80 \pm 0.85$ |
| Mixed | $\mathbf{51.67 \pm 1.61}$ | $50.89 \pm 2.87$ | $45.61 \pm 1.51$ |

Table 3: Accuracy (%) with Qwen2.5-7B Dir=Dirichlet, DT=Different Tokens, Single=Single-Token.

| Task | 15 Chains | | | 20 Chains | | |
|---|---|---|---|---|---|---|
| | Dir | DT | Single | Dir | DT | Single |
| Acre | $44.3 \pm 3.0$ | $\mathbf{47.1} \pm 3.2$ | $42.1 \pm 3.7$ | $42.7 \pm 1.6$ | $\mathbf{46.0} \pm 3.7$ | $38.2 \pm 1.6$ |
| Color Cube Rotation | $27.7 \pm 1.4$ | $\mathbf{28.0} \pm 2.6$ | $22.3 \pm 2.1$ | $13.2 \pm 0.7$ | $\mathbf{30.5} \pm 3.7$ | $22.3 \pm 2.8$ |
| Mixed | $32.5 \pm 0.6$ | $\mathbf{33.8} \pm 1.7$ | $23.0 \pm 1.4$ | $\mathbf{33.6} \pm 0.6$ | $30.6 \pm 0.6$ | $25.3 \pm 1.2$ |
| Number Sequence | $\mathbf{60.7} \pm 3.8$ | $50.7 \pm 10.3$ | $59.8 \pm 3.5$ | $58.3 \pm 6.8$ | $60.9 \pm 7.7$ | $\mathbf{62.2} \pm 4.9$ |
| Prime Factorization | $\mathbf{86.5} \pm 0.5$ | $84.1 \pm 5.1$ | $81.7 \pm 1.4$ | $\mathbf{85.3} \pm 0.2$ | $84.4 \pm 4.0$ | $81.5 \pm 3.0$ |

Table 4: Accuracy (%) $\pm$ standard deviation by task, number of chains, and method. Dir=Dirichlet, DT=Different Tokens, Single=Single-Token.

In Figure 5 we plot the evaluation accuracy across evaluation steps for different $k$ for the *different tokens* approach and for different temperatures (on different columns). Increasing $k$ to 4 helps improve the overall accuracy especially in acre and family relationships while doesn't improve in graph color and color-cube rotation. This can be intuitively understood by the result of Proposition 1 which claims that the choice of $k$ is task-sensitive for the *different token* method. For higher temperature it is evident that Dirichlet based sampling is the most robust to increasing temperatures (since the tokens used for sampling is not different but only the weights). The single-token and different tokens methods are more sensitive.

| Task | 5 Chains | | | 10 Chains | | |
|---|---|---|---|---|---|---|
| | Dir | DT | Single | Dir | DT | Single |
| Acre | **42.4** | 26.8 | 34.2 | **40.9** | 29.4 | 33.4 |
| Color Cube Rotation | **24.6** | 12.4 | 14.6 | **23.0** | 14.4 | 14.9 |
| Family Relationships | **38.6** | 33.6 | 19.4 | 42.4 | **48.3** | 28.6 |
| Graph Color | **23.8** | 5.8 | 9.6 | **28.4** | 5.0 | 15.7 |
| Number Sequence | **46.0** | 8.2 | 9.4 | **50.2** | 17.8 | 8.6 |

Table 5: Accuracy (%) with temperature T=1.2. Dir=Dirichlet, DT=Different Tokens, Single=Single-Token.

| Task | 5 Chains | | | 10 Chains | | |
|---|---|---|---|---|---|---|
| | Dir | DT | Single | Dir | DT | Single |
| Acre | **36.8** | 3.8 | 24.8 | **36.8** | 4.3 | 22.9 |
| Color Cube Rotation | **11.2** | 1.0 | 1.8 | **20.8** | 1.0 | 9.1 |
| Family Relationships | **17.8** | 2.0 | 0.6 | **36.9** | 7.1 | 6.2 |
| Graph Color | **13.2** | 0.2 | 0.2 | **15.1** | 0.5 | 0.8 |
| Number Sequence | **5.2** | 2.0 | 0.4 | **10.0** | 1.0 | 0.1 |

Table 6: Accuracy (%) with temperature T=1.6. Dir=Dirichlet, DT=Different Tokens, Single=Single-Token.

| Task | 5 Chains | | | 10 Chains | | |
|---|---|---|---|---|---|---|
| | Dir | DT | Single | Dir | DT | Single |
| Acre | **17.0** | 0.0 | 7.8 | **32.7** | 0.2 | 8.9 |
| Color Cube Rotation | **6.2** | 0.0 | 3.2 | **17.6** | 0.1 | 3.7 |
| Family Relationships | 2.4 | 0.0 | 2.4 | **28.1** | 0.2 | 2.0 |
| Graph Color | 0.8 | 0.0 | 0.0 | **5.2** | 0.0 | 0.0 |
| Number Sequence | 1.0 | 0.0 | 1.0 | 2.1 | 0.0 | 0.5 |

Table 7: Accuracy (%) with temperature T=2.0. Dir=Dirichlet, DT=Different Tokens, Single=Single-Token.

| Task | k=4 | k=8 |
|---|---|---|
| Acre | 47.4 | 48.6 |
| Color Cube Rotation | 22.2 | 26.0 |
| Family Relationships | 41.8 | 43.6 |
| Graph Color | 22.2 | 14.2 |
| Prime Factorization | 86.0 | 68.8 |

Table 8: Accuracy (%) by task, $k$, number of chains for the different tokens method.

| Task | k=2 | | k=4 | | k=8 | |
|---|---|---|---|---|---|---|
| | Dir | Single | Dir | - | Dir | - |
| GSM8K | **75.6** | 71.7 | 71.5 | - | 71.7 | - |
| MATH500 | **50.5** | 46.2 | 48.2 | - | 44.4 | - |

Table 9: Accuracy (%) by task and method for Math benchmarks. k=k, C=chains, Dir=Dirichlet, Single=Single-Token.

**Table 8 k-sensitivity (Different-Tokens).** Increasing the number of mixed tokens from $k = 4$ to $k = 8$ helps on relational/inductive tasks, *Acre* (47.4 to 48.6), *Color Cube Rotation* (22.2 to 26.0), and *Family Relationships* (41.8 to 43.6), but hurts more algorithmic ones, *Graph Color* (22.2 to 14.2)

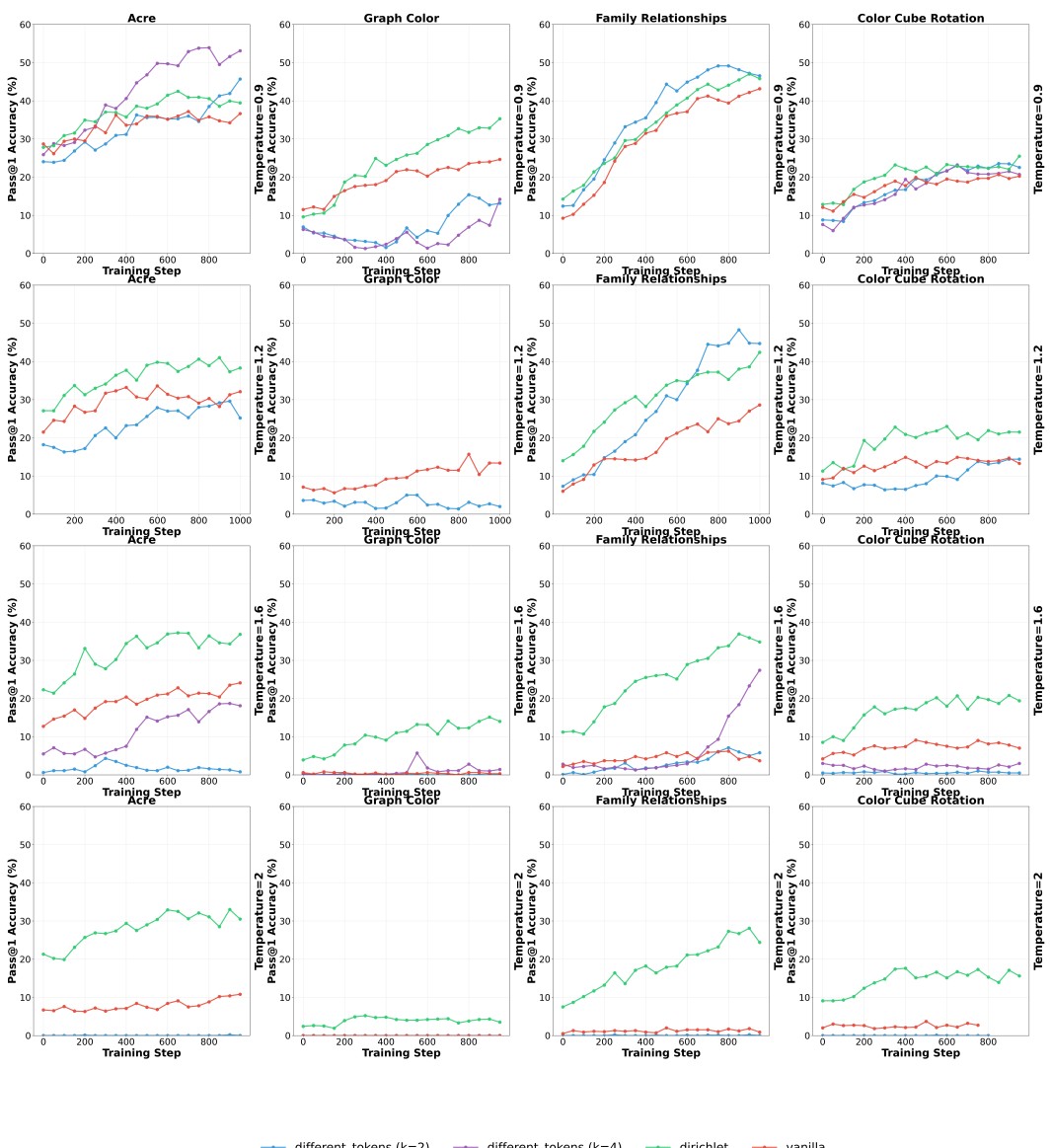

Figure 5: Progress of training for different algorithms under different temperatures (0.9 versus 1.2). Our methods can sometimes slow in the beginning (learning the mixture of tokens generation) but catch up quite quickly. The gains from the different-tokens technique is lesser with higher temperature possibly because of more irrelevant. Dirichlet based techniques are more robust to the temperature since top-$k$ tokens are the same, hence only the weights get more random with increasing temperature. We note that the different tokens based technique show improvement on some tasks with increasing $k$ on some tasks and Dirichlet based technique (not plotted) detoriates with increasing $k$, illustrating Proposition 1 that the choice of $k$ is an important task-dependent hyperparameter.

and *Prime Factorization* (86.0 to 68.8). This pattern underscores the predicted trade-off: larger $k$ broadens exploration but can dilute signal when precise token commitments matter, so a smaller $k$ is often safer for brittle, algorithmic settings.

**Table 9 Math benchmarks (Dirichlet vs. Single-Token).** On *GSM8K* and *MATH500*, Dirichlet mixing with a small set ($k = 2$) yields consistent gains over Single-Token decoding (75.6 vs. 71.7 on GSM8K; 50.5 vs. 46.2 on MATH500). As $k$ increases, accuracy declines (e.g., GSM8K 71.5 at $k = 4$; MATH500 48.2 at $k = 4$ and 44.4 at $k = 8$), suggesting that larger mixtures can over-soften the signal required for deterministic math. Dashes indicate settings not evaluated for the baseline. Together with Table 8, this supports the guidance that the optimal $k$ is task-dependent, with smaller $k$ favored for math-heavy tasks.

## B.1 BENCHMARKING EXTENSIONS OF MIXTURE OF TOKENS BASED GENERATION ON GSM-8K

This section presents the numerical experiments which illustrate efficacy of the different variations of the generalized mixture of tokens method. We do this for different small models of the Qwen family on GSM-8K (Cobbe et al., 2021) to a) see if different sampling and aggregation techniques affect the downstream performance b) which techniques form a good basis for selecting as generation mechanisms for our GRPO experiments.

Table 10: Performance of extensions of Training-Free Soft Thinking on GSM-8K. The sampling and aggregation methods have an impact on the downstream performance. For example, when doing element wise maximum (a method which is more sensitive to embeddings) gives 0 accuracy when used with top-k sampling but gives 44% with nucleus sampling.

| Sampling Method | Aggregation Method | Qwen3 1.7B | | | Qwen3 4B | | |
|---|---|---|---|---|---|---|---|
| | | k=2 | k=4 | k=8 | k=2 | k=4 | k=8 |
| *Baselines* | | | | | | | |
| Standard (Greedy) | — | | $75 \pm 3$ | | | $78 \pm 3$ | |
| Soft Thinking (Original) | — | | $77 \pm 3$ | | | $79 \pm 3$ | |
| *Other mixture of token methods* | | | | | | | |
| Top-k | Prob. Weighted Avg. | $76 \pm 3$ | $75 \pm 3$ | $76 \pm 3$ | $80 \pm 3$ | $79 \pm 3$ | $79 \pm 3$ |
| Top-k | Dirichlet Weighted Avg. | $80 \pm 3$ | $80 \pm 3$ | $80 \pm 3$ | $79 \pm 3$ | $75 \pm 3$ | $80 \pm 3$ |
| Top-k | Element Wise Maximum | $3 \pm 1$ | 0 | 0 | 0 | 0 | 0 |
| Min-p | Prob. Weighted Avg. | | $78 \pm 2$ | | | $81 \pm 2$ | |
| Min-p | Dirichlet Weighted Avg. | | $81 \pm 2$ | | | $83 \pm 2$ | |
| Min-p | Element Wise Maximum | | 0 | | | 3 | |
| Nucleus Sampling | Prob. Weighted Avg. | | $78 \pm 2$ | | | $78 \pm 2$ | |
| Nucleus Sampling | Dirichlet Weighted Avg. | | $78 \pm 2$ | | | $77 \pm 2$ | |
| Nucleus Sampling | Element Wise Maximum | | $18 \pm 2$ | | | $44 \pm 2$ | |
| k-sampling | Prob. Weighted Avg. | $74 \pm 3$ | $78 \pm 2$ | $78 \pm 3$ | $78 \pm 3$ | $79 \pm 3$ | $79 \pm 3$ |
| k-sampling | Dirichlet Weighted Avg. | $79 \pm 3$ | $79 \pm 3$ | $85 \pm 3$ | $79 \pm 3$ | $76 \pm 3$ | ? |
| k-sampling | Element Wise Maximum | $3 \pm 1$ | 0 | 0 | 5 | 5 | 0 |

# C  PROOFS

## C.1  PROOF OF PROPOSITION 1

*Proof.* Let $p_t = h(\mathbf{X}_{t-1})$ denote the probability distribution over dictionary $\mathcal{D}$ at step $t$. For each rollout $g \in [G]$, we sample $k$ tokens $S_g = \{z_{g,1}, \ldots, z_{g,k}\}$ without replacement.

**Part 1:** For the expected number of unique tokens $L_t$ across $G$ trajectories, let $q_j(k)$ denote the probability that token $j$ is included when sampling $k$ tokens. For sampling without replacement:

$$q_j(k) = 1 - \prod_{i=0}^{k-1} \frac{1 - p_j - \sum_{m \in S_i} p_m}{1 - \sum_{m \in S_i} p_m} \geq q_j(k-1)$$

where $S_i$ denotes the first $i$ sampled tokens. Since $q_j(k)$ increases with $k$, the expected number of unique tokens can be computed using indicator variables. Let $I_j = 1$ if token $j$ appears in at least

one rollout, 0 otherwise. Then $L_t = \sum_{j=1}^{|\mathcal{D}|} I_j$ and by linearity of expectation:

$$\mathbb{E}[L_t] = \sum_{j=1}^{|\mathcal{D}|} \mathbb{P}(I_j = 1) = \sum_{j=1}^{|\mathcal{D}|} \left[ 1 - (1 - q_j(k))^G \right]$$

is monotonically increasing in $k$.

**Part 2:** Let $Z \sim p$ and write $Y := e_Z$. For any set $S \subseteq D$ with $p(S) := \sum_{j \in S} p_j > 0$, define the conditional mean

$$m(S) := \mathbb{E}[Y \mid Z \in S] = \sum_{j \in S} \frac{p_j}{p(S)} e_j.$$

Under the standard sequential PPS-without-replacement construction, we can couple the random sampled sets so that $S_k \subset S_{k+1}$ almost surely. In this scheme, one rollout's mixture embedding with sample size $k$ is exactly

$$X_k = m(S_k).$$

Let $\mu := \mathbb{E}Y = \sum_{j \in D} p_j e_j$. For any random vector $U$ with an independent copy $U'$,

$$\mathbb{E}\|U - U'\|^2 = 2\operatorname{tr}\big(\operatorname{Cov}(U)\big) = 2\big(\mathbb{E}\|U\|^2 - \|\mathbb{E}U\|^2\big). \tag{1}$$

Hence it suffices to prove that

$$\operatorname{tr}\big(\operatorname{Cov}(X_{k+1})\big) \leq \operatorname{tr}\big(\operatorname{Cov}(X_k)\big). \tag{2}$$

Fix a realization of the nested sets $S_k \subset S_{k+1}$. Consider the two-atom $\sigma$-algebras $\mathcal{G}_k := \sigma(\{Z \in S_k\})$ and $\mathcal{G}_{k+1} := \sigma(\{Z \in S_{k+1}\})$. By construction, conditioning on $\{Z \in S_{k+1}\}$ is strictly *coarser* information than conditioning on $\{Z \in S_k\}$ (because $S_k \subset S_{k+1}$). Since $m(S) = \mathbb{E}[Y \mid \sigma(\{Z \in S\})]$, the variance of these $L^2$-projections is monotone in the information:

$$\operatorname{Var}(\mathbb{E}[Y \mid \mathcal{G}_{k+1}]) \leq \operatorname{Var}(\mathbb{E}[Y \mid \mathcal{G}_k]). \tag{3}$$

(Equation (3) is the standard monotonicity of $\operatorname{Var}(\mathbb{E}[Y \mid \mathcal{H}])$ with respect to the partial order of $\sigma$-algebras: coarser $\mathcal{H}$ yields smaller variance.)

Taking expectation over the randomness of the sampled sets yields

$$\mathbb{E}\operatorname{tr}\big(\operatorname{Cov}(X_{k+1})\big) = \mathbb{E}\operatorname{tr}\Big(\operatorname{Cov}\big(\mathbb{E}[Y \mid \mathcal{G}_{k+1}]\big)\Big) \leq \mathbb{E}\operatorname{tr}\Big(\operatorname{Cov}\big(\mathbb{E}[Y \mid \mathcal{G}_k]\big)\Big) = \mathbb{E}\operatorname{tr}\big(\operatorname{Cov}(X_k)\big),$$

which is (2). Combining (2) with the identity (1) for $U = X_k$ and $U' = X_k'$ gives the claim. $\qquad\square$

## C.2 Background on Dirichlet distribution

The Dirichlet distribution $\operatorname{Dir}(\boldsymbol{\alpha})$ is defined on the $(K-1)$-simplex $\Delta^{K-1} = \{\mathbf{x} \in \mathbb{R}^K : x_i \geq 0, \sum_i x_i = 1\}$ with density,

$$f(\mathbf{x} \mid \boldsymbol{\alpha}) = \frac{1}{B(\boldsymbol{\alpha})} \prod_{i=1}^{K} x_i^{\alpha_i - 1}, \qquad B(\boldsymbol{\alpha}) = \frac{\prod_{i=1}^{K} \Gamma(\alpha_i)}{\Gamma\big(\sum_{i=1}^{K} \alpha_i\big)}.$$

where $\Gamma$ is the gamma function. Parameterizing as $\boldsymbol{\alpha} = c\,\mathbf{p}$ with concentration $c > 0$ and base probabilities $\mathbf{p} \in \Delta^{K-1}$ gives $\mathbb{E}[X_i] = p_i$, $\operatorname{Var}(X_i) = \frac{p_i(1-p_i)}{c+1}$, and $\operatorname{Cov}(X_i, X_j) = -\frac{p_i p_j}{c+1}$ for $i \neq j$. Thus $c$ controls spread (smaller $c \Rightarrow$ broader draws) while $\mathbf{p}$ sets the center. Figure 6 shows $K = 3$ cases at $c = 1$: a symmetric center ($\mathbf{p} = [\frac{1}{3}, \frac{1}{3}, \frac{1}{3}]$), an asymmetric middle ($[0.6, 0.3, 0.1]$), and a corner ($[0.95, 0.04, 0.01]$). As the center approaches a vertex, samples concentrate near that corner, illustrating the intuition that "points in the corner stay in the corner."

# D  Related Work on Other Test-Time Scaling Methods in LLMs

*Other Test-Time Scaling Methods in LLMs:* Yao et al. (2024) combines fast and slow thinking modes and switches between them based on instance difficulty, so the model spends extra steps only when useful. Zhang et al. (2025a) provides a single view that covers many prior "more steps help"

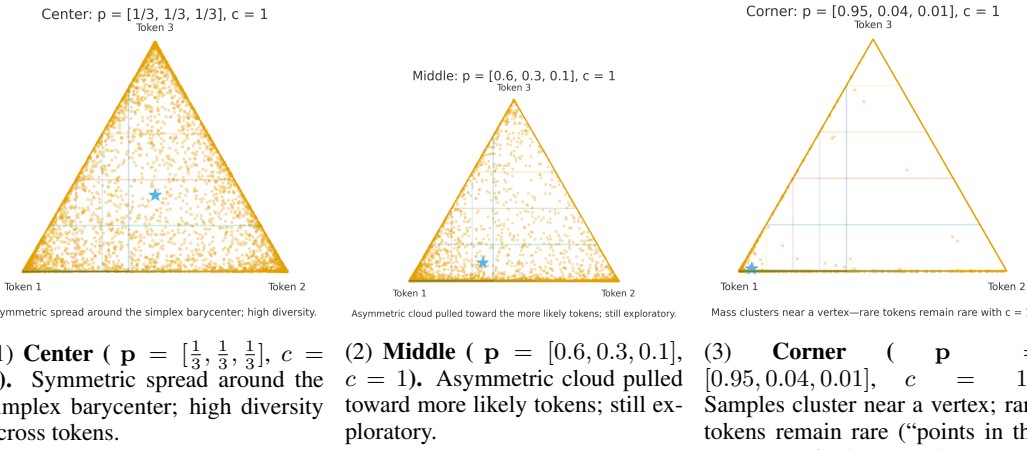

(1) **Center ( $\mathbf{p} = [\frac{1}{3}, \frac{1}{3}, \frac{1}{3}]$, $c = 1$).** Symmetric spread around the simplex barycenter; high diversity across tokens.

(2) **Middle ( $\mathbf{p} = [0.6, 0.3, 0.1]$, $c = 1$).** Asymmetric cloud pulled toward more likely tokens; still exploratory.

(3) **Corner ( $\mathbf{p} = [0.95, 0.04, 0.01]$, $c = 1$).** Samples cluster near a vertex; rare tokens remain rare ("points in the corner stay in the corner").

Figure 6: **Dirichlet perturbations on the probability simplex (all with $c = 1$).** Each panel shows 4000 samples from $\mathrm{Dir}(c\,\mathbf{p})$ with a different center $\mathbf{p}$; the star marks the base distribution. Lower $c$ yields broader, more exploratory draws; the choice of $\mathbf{p}$ determines where mass concentrates.

methods, and lets the system control how much slow vs fast reasoning to use in a fine-grained way. Ji et al. (2025) introduce DEBATER, a retriever that learns better document embeddings by running a simple step-by-step thinking process, giving stronger retrieval when more test-time thinking is allowed. (Yao et al., 2023) Tree of Thoughts scales by exploring branches of partial thoughts, tuning breadth and depth to trade off cost and accuracy. Geiping et al. (2025) scales compute by repeating a recurrent block to any depth at inference, growing inner iterations rather than only producing more tokens. Xu et al. (2024) LaRS learns a latent "reasoning skill" space and a policy that picks the skill level for a question, then chooses in-context examples that match it, which adapts test-time effort to the task. Wang et al. (2025) improves RLVR by updating only on forking tokens, the points where the path can split, which focuses extra samples where they matter most. Hou et al. (2025) pretrains with trial-and-error plus self-check CoT, then scales RL by over-sampling to raise sample diversity, which supports stronger test-time sampling without training collapse. Zhao et al. (2025) uses self-play to generate diverse trajectories at inference, which can be ramped up for harder inputs. One can similarly extend our techniques to other recent algorithms including GSPO and DR-GRPO.

# E EXAMPLE PROBLEMS

For MoT-G methods we only show the token with the highest probability.

---

**Question**

You are a researcher studying causal relationships using Blicket experiments. In these experiments, certain objects (called 'blickets') have the hidden property of activating a detector, causing its light to turn on.

Each example shows the results of placing different combinations of objects on the detector. Each object is described by color, material and shape. Your task is to determine whether a new combination of objects will cause the detector to activate.

After observing the previous examples, respond with: - "on" if you can determine the detector light will turn on - "off" if you can determine the detector light will stay off - "undetermined" if there is insufficient evidence to reach a conclusion

Do not use quotation marks in your answer.

Previous experimental results: brown rubber cube → off gray rubber cylinder → on gray rubber cylinder, brown rubber cube → on blue metal cylinder, blue rubber cube, purple rubber cube, brown metal cube, gray metal cube, red rubber cylinder → on blue rubber cube, purple rubber cube, red rubber cylinder → on brown metal cube, red rubber cylinder, gray metal cube → off

New test case: blue rubber cube

What is the detector light status?

---

**Answer from Standard Generation**

Let's analyze the pattern and determine the effect of the new test case.⟨/think⟩

1) First, we identify the objects that cause the detector to turn on: - Gray rubber cylinder → on - Gray rubber cube, brown rubber cube → on - Blue metal cylinder, blue rubber cube, purple rubber cube, brown metal cube, gray metal cube, red rubber cylinder → on - Blue rubber cube, purple rubber cube, red rubber cylinder → on - Black rubber cube → on

2) Observing these combinations, we notice: - The combination with a cylinder almost always causes the detector to turn on (Gray rubber cylinder, Blue metal cylinder, Blue rubber cube, Red rubber cylinder). - The objects causing the detector to turn on are almost always cubes or metal cubes. - The color and shape of the objects don't seem to have a noticeable impact on whether the detector turns on.

3) Given these observations, while we can't exclude the possibility of any black rubber cubes or metal cubes with other colors, the sequence strongly suggests that cubes will lead to an activation.

4) The new test case is a blue rubber cube.

5) Based on the pattern observed from previous test cases, we can infer that a blue rubber cube will likely activate the detector.

⟨answer⟩on⟨/answer⟩

---

To determine whether the detector will activate with the new test case of blue rubber cube, we need to analyze the previous results to identify any consistent patterns. Let1. The first test case ( brown rubber cube, off indicates that a brown rubber cube alone does not cause the detector to activate. 2. The second test case (gray rubber cylinder → on) shows that a gray rubber cylinder alone causes the detector to activate. 3. The third test case (gray rubber cylinder, brown rubber cube → on) shows that both a gray rubber cylinder and a brown rubber cube can cause activation detector to activate. . 4. The fourth test case (blue metal cylinder, blue rubber cube, purple rubber cube, brown metal cube, gray metal cube, red rubber cylinder → on) shows that a combination of a blue metal cylinder and a blue rubber cube, cause the detector to activate. 5. The fifth test case (blue rubber cube, purple rubber cube, red rubber cylinder → on) indicates that a combination of a blue rubber cube, a purple rubber cube can cause the detector to activate. 6. The sixth test case (brown metal cube, red rubber cylinder, gray metal cube → off) shows that a combination of a brown metal cube, a red rubber cylinder does not cause the detector to activate.

From these results: - A single brown (brown rubber cube) does not cause the detector to activate. - A single object (gray rubber cylinder) causes the detector to activate. - A combination of objects (gray rubber cylinder, brown rubber cube) causes the detector to activate. - A combination of objects (blue metal cylinder, blue rubber cube, purple rubber cube, brown metal cube, gray metal cube, red rubber cylinder) causes the detector to activate. - A combination of objects (blue rubber cube, purple rubber cube, red rubber cylinder) causes the detector to activate. - A single object (brown metal cube) does not cause the detector to activate. - A single object (red rubber cylinder) does not cause the detector to activate. - A combination of objects (brown metal cube, red rubber cylinder) gray metal cube) does not cause the detector to activate.

Based on these observations: - A combination of objects containing a gray rubber cylinder will cause the detector to activate. - The presence of a brown rubber cube does not cause the detector to activate. - The presence of a blue rubber cube, a rubber cube, and red rubber cylinder will cause the detector to activate. - A combination of objects containing a blue rubber cube and a gray rubber cylinder does not cause the detector to activate.

The new test case is a blue rubber cube. From our analysis, we can conclude that a combination of a gray rubber cylinder and a blue rubber cube will cause the detector to activate. Since the blue rubber cube is not a combination, it is not clear whether it will activate the detector. Therefore, since a blue rubber cube alone does not cause the detector to activate, it is less likely that it will cause it.Therefore, the detector light will most likely turn on with the new test case of blue rubber cube. ⟨/think⟩

⟨answer⟩on⟨/answer⟩

**Answer from Different Tokens MoT-G**

Let's break the previous examples results to see the the detector will turn based a new test case of placing blue rubber cube.

1. FromFirst test Case**: Brown rubber cube ( Off) - This detector rubber cube is different from the rubber rubber cube in but turned on. This, this alone does not provide enough evidence to conclude that outcome for blue blue rubber cube alone its own.

2. **Second Test Case:** Gray rubber cylinder (on) - The blue rubber cube, placed on with the gray rubber cylinder. The this test, the blue turned on,

suggests a at presence of a cylinder rubber cylinder might trigger on detector on,3. **Third Test Cases:** Gray rubber cube, blue rubber cube →on) - The blue rubber cube was placed next to the gray rubber cylinder, brown rubber cube. The detector turned on again. This suggests that the presence of both the gray rubber cylinder and a brown rubber cube can cause the detector.

4. **Fourth Test Case:** Blue metal cylinder, blue rubber cube, purple rubber cube, brown metal cube, gray metal cube, red rubber cylinder →on) - The blue rubber cylinder was placed with to the blue rubber cube, purple rubber cube, brown metal cube, gray metal cube, and red rubber cylinder. The detector turned on. This suggests that the presence of multiple metals and non objects can activate the detector.

5. **Fifth Test Case:** Blue rubber cube, purple rubber cube, red rubber cylinder (on) - The blue rubber cube and placed next to the purple rubber cube and red rubber cylinder. The detector turned on. This indicates that the presence of multiple least two different cubes might activate the detector.

6. **Sixth Test Case:** Brown metal cube, red rubber cylinder, gray metal cube (off) - The brown metal cube was placed next to the red rubber cylinder, gray metal cube. The detector turned not turn on, This suggests that the presence of the metal cube might not always turn the detector,Based these results,- The blue of the gray cylinder cylinder (on) suggests that the presence can turn activated by- The presence of a metal cubes rubber cubes (on) also that the detector can also activated. - The absence of a single blue rubber cube does not necessarily sufficient information to determine the activation's activation statusTherefore, based on the pattern results, we detector is will is the new test case ( a blue rubber cube is ⟨offetermined** because There are not evidence to conclude whether it will turn on or not off.

## E.2   COLOR CUBE ROTATION

**Question**

A cube has: - a silver top side - a cyan right side - a magenta front side - a orange left side - a blue back side - a pink bottom side
The cube is rotated so that the side which was before at the left is now at the top.
What is now the color of the left side of the cube? Provide only the color as your final answer.

To solve this puzzle, let's break it down step by step:
1. **Identify the Initial Configuration:** - The cube has sides labeled according to their colors: a silver top side, a cyan right side, a magenta front side, an orange left side, a blue back side, and a pink bottom side.
2. **Understand the Rotation:** - The problem states that the side which was on the left side is now on top.
3. **Logical Reasoning:** - Since the side that was on the left is now on top, we can deduce that the side that was originally on the left must have been the one that moves to the top after the rotation. - This implies that the side originally on the left must have been the side that had to move to become the top side.
4. **Determine the Moved Side:** - The side that must have moved is the one that was originally on the left. This side was not the top one, which is silver.
5. **Final Configuration:** - Now that this is understood, we can see that the side that was originally on the left side, which is the side that had to move, is now where the silver side was originally.
6. **Conclusion:** - Therefore, the side that was originally to the left, which is now on top, has to be the side that is now the left side.
So, the original pink bottom side is now the left side of the cube.
⟨/think⟩
⟨answer⟩orange⟨/answer⟩

LetTo determine the color of the left side after the rotation, we need need to understand how the sides of a cube change when it's rotated. A cube has six faces, and each face can be rotated by its position relative to the others. When a cube is rotated, the face that was on the bottom moves to the top, the face that was on the front moves to the back, the face that was on the back moves to the left, the face on was on the back moves to the front, and the face that was on the left moves to the bottom. Given the initial positions of the sides: - The silver side is at the top. - The cyan side is at the right - The magenta side is at the front - The orange side is at the left - The blue side is at the back - The pink side is at the bottom
When the cube is rotated so that the side that was at the left is now at the top, the left side will now be at the bottom. the all bottom side will now be at the top. Therefore side that was at the bottom will now be the new that was originally at the front, Therefore, the side that was originally at the front, which is magenta, will now be at the bottom. The side that was originally at the back, which is blue, will now be at the front. The side that was originally at the right, which is cyan, will now be at the right. The side that was originally at the left, which is orange, will now be at the back. The side that was originally at the top, which is silver, will now be at the front. The side that was originally at the bottom, which is pink, will now be at the back.
Therefore, after color of the left side of the cube after magenta. ⟨/think⟩
⟨answer⟩magenta⟨/answer⟩

> ### Answer from Different Tokens MoT-G
>
> First determine the color of the left side of the cube after it is been rotated, let need to consider how orientation configuration and the effect of the rotation. each cube. the cube. TheFirst. **Ident Configuration:** - Top cube side is silver. - The right side is cyan. - The front side is magenta. - The left side is orange. - The back side is blue. - The bottom side is pink.
>
> 2. **Rotation Analysis**:** - The side which was initially on the left ( now at the top. This means the side side has moved from the top.
>
> . Now. **Effect of the: S Sides:** - Rotation a side is rotated, the sides that was at on one bottom side to the top. The implies the left that was initially at will now at the top position4. **D Configuration:** - The side which was initially at the left isorange) is now in the top. Therefore, the cube, the side of the left side of the cube is **orange**.⟨/think⟩ ⟨answer⟩Orange⟨/answer⟩

## E.3   FAMILY RELATIONSHIPS

> ### Question
>
> Phoenix is married to Eleanor. They have a child called Zion. Zion is married to Margaret. They have children called Thomas and Henry. Michael is married to Isabella. They have a child called Margaret.
> What relation is Zion to Isabella? Answer with a single word.

> ### Answer from Standard Generation
>
> First, we'll see if there's any direct information about Zion and Isabella's relationship. Then, we can use the relationships among the rest of the characters to deduce the relationship between Zion and Isabella. Let's consider these steps:⟨/think⟩
> ⟨answer⟩mother⟨/answer⟩

> ### Answer from Dirichlet MoT-G
>
> To determine the relationship between Zion and Isabella, we can trace the lineage from Zion back to Michael and as Isabella is his wife. Zion is the father of Thomas, who is the daughter of Michael. Isabella. Therefore, Zion is the father of Isabella's child. ⟨/think⟩ ⟨answer⟩ father⟨/answer⟩

> ### Answer from Different Tokens MoT-G
>
> Z determine Zion relation between Zion and Isabella, we's analyze the given relationships step:
> the information: . We know given that Phoenix is married to Margaret.2. We are also told that Margaret is married to Isabella. 3. Since the above piece of information ( we know that Zion and Margaret are married.
> 4. Since, Zion is the father of Margaret. and makes Zion is Isabella's husband.
> Basedthink⟩ ⟨answer⟩ husband ⟨/answer⟩

## E.4   GRAPH COLOR

> ### Question
>
> Please provide a coloring for this graph such that every vertex is not connected to a vertex of the same color. The graph has these properties:
> Vertices: [0, 1, 2, 3, 4, 5, 6, 7, 8, 9] Edges: [(0, 2), (0, 5), (3, 4)] Possible colors: [1, 2, 3]
> Return your solution as a JSON map of vertices to colors. (For example: \"0": 1, "1": 2, "2": 3\.)

To solve this coloring problem, I need to ensure that every vertex is not connected to a vertex of the same color. The graph has the following properties: Vertices: [0, 1, 2, 3, 4, 5, 6, 7, 8, 9] Edges: [(0, 2), (0, 5), (3, 4)] Possible colors: [1, 2, 3]

First, let's consider the edges: - Edge from vertex 0 to vertex 2 - Edge from vertex 0 to vertex 5 - Edge from vertex 3 to vertex 4

We must color vertices with different colors to ensure no two connected vertices share the same color. Let's start with vertex 2 and vertex 5: - Color vertex 2 with color 1 - Color vertex 5 with color 2

Next, we examine the remaining vertices and edges: - Edge from vertex 3 to vertex 4: both vertices must have different colors - Vertices 6 to 9 have no edges yet.

Since vertices 6 to 9 haven't been colored yet, let's color vertex 6 with color 1 and vertex 7 with color 2: - Color vertex 6 with color 1 - Color vertex 7 with color 2

Now we can color vertex 8 with color 3: - Color vertex 8 with color 3

Finally, let's color vertex 9 with color 1: - Color vertex 9 with color 1

We have ensured that no two connected vertices share the same color. The final coloring can be represented as: \"0": 1, "1": 2, "2": 1, "3": 2, "4": 3, "5": 2, "6": 1, "7": 2, "8": 3, "9": 1\ ⟨/think⟩ ⟨answer⟩\"0": 1, "1": 2, "2": 1, "3": 2, "4": 3, "5": 2, "6": 1, "7": 2, "8": 3, "9": 1\⟨/answer⟩

Let's start by considering the structure graph and the constraints. The vertices are connected as follows: - Vertex 0 is connected to Vertex 2 and 5. - Vertex 3 is connected to vertices 4. - We need to ensure that no two connected vertices have the same color.

Given the possible colors are1, 2, 3], let's try to color the vertices step by step:

1. **Color Vertex 0:** Since 0 is connected to vertices2 and 5, we can color it0 with color 1 ( This ensures vertex2 and 5 are not be colored with 1.

2. **Color Vertex 2:** Vertex 2 is connected to 0, so it can be colored with color 2.3. **Color Vertex 5:** Vertex 5 is connected to 0, so it can be colored with color 2.

4. **Color Vertex 3:** Vertex 3 is connected to vertex 4, so we can be colored with color 1.

5. **Color Vertex 4:** Vertex 4 is not connected connected to any other vertex yet so it can be colored with color 3. This ensures that vertex 4 does not connected to vertex 0, which is colored with color 1.

6. **Color Vertex 6:** Vertex 6 is not directly connected to any other vertex, so it can be colored with color 3.

7. **Color Vertex 7:** Vertex 7 is not to vertex 6 ( so it can be colored with color 3.

8. **Color Vertex 8:** Vertex 8 is not directly connected to any other vertex, so it can be colored with color 3.

9. **Color Vertex 9:** Vertex 9 is not directly connected to any other vertex, so it can be colored with color 3.

So final coloring is is as\"0": 1, "1": 2, "2": 2, "3": 1, "4": 3, "5": 2, "6": 3, "7": 3, "8": 3, "9": 3\

⟨/think⟩ ⟨answer⟩\"0": 1, "1": 2, "2": 2, "3": 1, "4": 3, "5": 2, "6": 3, "7": 3, "8": 3, "9": 3\⟨/answer⟩

LetTo solve this coloring, we need to color that every vertex adjacent vertices share the same color. Let given graph has vertices \0, 1, 2, 3, 4, 5, 6, 7, 8, 9\] and edges [(0, 2), (0, 5), (3, 4)] The available colors are [1, 2, 3].

's try by analyzing the connections to determine the constraints:
1. Vertex 0 is connected to vertices 2 and 5. Since vertex2 and 5 have not already to we can both the color. 2's choose color 1 to vertex 0 and2. Vertex 2 is also connected to vertex 5. Since vertex0 is already yet assigned, we need assign it 2 to vertex 3. 3. Now 0 is connected to vertex 4. Since vertex4 is not yet colored, we can assign color 1 to vertex 4. 4. Now 2 is connected to vertex 3. Since 5 is already colored, we need to assign color new color to vertex 4. Let's assign color 1 to vertex 4. 5. Vertex 5 is connected to vertex 0 and Since 0 is already colored with we need to assign a different color to vertex 5. Let's assign color 3 to vertex 5.

6. Finally 6, 7, and8, and 9 are not connected to any other yet so they can be any color. We's assign colors 1 to vertex 6, color 2 to vertex 7, color 3 to vertex 8, and color 1 to vertex 9. Therefore, the final is be: \"0": 1, "1": 2, "2": 1, "3": 2, "4": 1, "5": 3, "6": 1, "7": 2, "8": 3, "9": 1\⟨/think⟩

## E.5 PRIME FACTORIZATION

Find the prime factorization of 5. Write the factors separated by × (Example: for 12 the answer would be: $2 \times 2 \times 3$)

Let's analyze the number 5 to find its prime factors. A prime number is a number greater than 1 that has no divisors other than 1 and itself. Let's start by testing if 5 is divisible by any number less than its square root (approximately 2.24), which means we only need to check up to 5. 5 is not divisible by any of these numbers. Therefore, 5 itself is a prime number. ⟨/think⟩
⟨answer⟩5⟨/answer⟩

ToTo find the prime factorization of 5, we need to determine which prime can divide multiplied together to give 5. Since 5 is a prime number, it has only two positive divisor: which is itself. Therefore, the prime factorization of 5 is simply 5. This ⟨/think⟩ ⟨answer⟩5⟨/answer⟩

To find the prime factorization of 5, we need to determine its prime numbers multiply together to equal 5. The know that 5 is a prime number itself, so it has no divisors other than 1 and itself. Therefore, the prime factorization of 5 is just 5. However prime of by × are: 5 × 1. ⟨/think⟩ ⟨answer⟩ 5 × 1 ⟨/answer⟩

## E.6 SELF REFERENCE

Given the truthfulness of these statements, please tell me the number of possible solutions: - Statement 1: 'At least 6 of these 7 statements are true.' - Statement 2: 'At most 3 of these 7 statements are false.' - Statement 3: 'Exactly 0 of these 7 statements are true.' - Statement 4: 'Exactly 2 of these 7 statements are false.' - Statement 5: 'Either Statement 3 or Statement 4 is true, but not both.' - Statement 6: 'The number of true statements is a prime number.' - Statement 7: 'The number of false statements is a composite number.'

Let's approach this step-by-step:⟨/think⟩ ⟨answer⟩3⟨/answer⟩

Let's approach this step by step:⟨/think⟩ 1. First, consider Statement 1, which says that at least 6 of the statements are true.

2. If Statement 1 is true, then Statements 2, 3, 4, 5, 6, and 7 would need to include 6 and possibly another true statement. Let's explore this further.

3. If Statement 3 is true, then exactly 0 of the statements are true. This contradicts Statement 1, as it would mean only 0 statements are true, which doesn't make sense as Statement 1 already stated that at least 6 are true. Therefore, Statement 3 must be false, meaning there is at least 1 true statement.

4. If Statement 3 is false, then there must be at least 1 true statement (because Statement 3 is false).

5. Now consider Statement 2. It claims that at most 3 of the 7 statements are false.

6. If Statement 2 is true, then at most 3 statements are false. This suggests there could be 1, 2, or 3 false statements.

7. However, statements 3 and 4 claim that there is at least 1 true statement, contradicting Statement 2. So we discard statement 2 as it leads to a contradiction.

8. With Statement 2 ruled out, let's consider the remaining statements.

9. Statement 5 claims that either Statement 3 is true and Statement 4 is false, or vice versa (but not both). Since we know Statement 3 is false, Statement 4 must be true.

10. This means there is 1 true statement: Statement 4.

11. With Statement 4 true, Statements 5, 6, and 7 must be false. Now, we need to check if the number of true statements is a prime number, which 1 does indeed satisfy.

12. Statement 6 claims that the number of true statements is a prime number, which is true since we've established there is exactly 1 true statement.

13. Statement 7 claims that the number of false statements is a composite number. There is exactly 1 false statement (Statement 4), which is a composite number.

14. Therefore, there is no contradiction between the statements. So, we have exactly 3 possible solutions.
⟨answer⟩3⟨/answer⟩

Let's break by considering the 3. "At least 6 of these 7 statements are true." If means is there there least 6 of of the 7 statements are true. but is the 7 mustThe number of false statements is a composite number") must be false. Therefore, Statement 1 cannot be true, false. Since Statement Statement 7 must false.

Nowthink⟩ ⟨ththink⟩Now, consider's look Statement 2: 'At most 3 of these 7 statements are false." Since we 7 is false, Statement most 3 of the statements7 statements can be true, However 1 says at at least 6 of are true, so means Statement can at7 or statement and. This means with 4 false statements. Therefore Statement Statement 2 is true. ⟨/think⟩

⟨think⟩Next, let's consider Statement 3: "Exactly 0 of these 7 statements are true." Since we've established that Statement 7 is false, Statement 3 must be false. So, Statement 3 is false. ⟨/think⟩

⟨think⟩Now, let's consider Statement 4: "Exactly 2 of these 7 statements are false." Since we've established that the number of false statements is 3, Statement 4 must be false. So, Statement 4 is false. ⟨/think⟩

⟨think⟩Next, let's consider Statement 5: "Either Statement 3 or Statement 4 is true, but not both." Since we've established that Statements 3 and 4 are false, Statement 5 is true. ⟨/think⟩

⟨think⟩Now, let's consider Statement 6: "The number of true statements is a prime number." Since both Statement 3 (false) and Statement 4 (false) are contradicting at least one statement, Statement 6 must be false. So, Statement 6 is false. ⟨/think⟩

⟨think⟩Now, the only possible numbers of true statements are 1 and 3 (based on the established facts). Statement 6 being false means the number of true statements could be 3 (prime). Statement 5 being true means the number of true statements could also be 1, which contradicts Statement 6. Therefore, the only possibility left is that there are 3 true statements. ⟨/think⟩

⟨answer⟩3⟨/answer⟩

Therefore, there is only one possible solution: there are 3 true statements.

## E.7 SIMPLE INTEGRATION

Calculate the antiderivative: $\int 50*X**9 + 4*X$ dx When performing calculations, please follow these guidelines: 1. Use ** instead of \textasciicircum\\ to represent exponents. For example, write 7*X**2 instead of 7*X\textasciicircum\\2. 2. Always include the * symbol for all multiplication operations in your reasoning steps. For example, write '-3*X**3*sin(X) - 9*X**2*cos(X) + 18*X*sin(X) + 18*cos(X) + C' instead of '-3x3sin(x) - 9x2cos(x) + 18xsin(x) + 18cos(x) + C'.

⟨underline⟩ Step 1: Identify the integrand, which is the expression to be integrated.⟨/underline⟩ ⟨ul⟩ ⟨li⟩The integrand is 50*X**9 + 4*X⟨/li⟩ ⟨/ul⟩ ⟨ul⟩ ⟨li⟩Remember the formula for an integral of a sum is the sum of the integrals: $\int (f(X) + g(X))$ dX = $\int f(X)$ dX + $\int g(X)$ dX where $\int f(X)$ dX is the integral of f(X) and $\int g(X)$ dX is the integral of g(X).⟨/li⟩ ⟨/ul⟩

⟨ul⟩ ⟨li⟩Next, we need to integrate each term separately.⟨/li⟩ ⟨/ul⟩

⟨ul⟩ ⟨li⟩For the first term, 50*X**9, apply the power rule of integration: $\int X$\textasciicircum\\n dX = (X\textasciicircum\\(n+1))/(n+1) + C⟨/li⟩ ⟨/ul⟩

⟨ul⟩ ⟨li⟩For the second term, 4*X, integrate directly since it's a linear function:⟨/li⟩ ⟨/ul⟩

⟨ul⟩ ⟨li⟩$\int 4*X$ dX = 4/2 * $\int X$ dX = 2*X + C⟨/li⟩ ⟨/ul⟩

⟨ul⟩ ⟨li⟩So the antiderivative of 50*X**9 + 4*X is: ⟨/li⟩ ⟨/ul⟩

⟨ul⟩ ⟨li⟩50*(X**10)/10 + 2*X + C⟨/li⟩ ⟨/ul⟩

⟨ul⟩ ⟨li⟩Which simplifies to: ⟨/li⟩ ⟨/ul⟩

⟨ul⟩ ⟨li⟩5*X**10 + 2*X + C⟨/li⟩ ⟨/ul⟩ ⟨/think⟩

⟨answer⟩ The final answer is 5*X**10 + 2*X + C. ⟨/answer⟩

---

**Answer from Dirichlet MoT-G**

Let's break down the calculation step-by-step.
1. ** integral we need to solve is ∫ 50*X**9 + 4*X dx 2. We can split this integral into two parts: ∫ 50*X**9 dx + ∫ 4*X dx
3. For the first part ∫ 50*X**9 dx, we use use out the constant 50: 50 * ∫ X**9 dx - The antiderivative of X**n (where n ≠ a constant) is X**(n+1)/(n+1, so: 50 * X**(10 / 10 5*X**10 5. For the second part, ∫ 4*X dx: the antiderivative of X is X: 4 * X**2 / 2 = 2 ***2 6. Combining both parts, we get: 5*X**10 + 2*X**2 7. Adding, we add the constant of integration C: 5*X**10 + 2*X**2 + C Therefore, the antiderivative of 50*X**9 + 4*X dx is 5*X**10 + 2*X**2 + C. ⟨/think⟩ ⟨answer⟩5*X**10 + 2*X**2 + C⟨/answer⟩

---

**Answer from Different Tokens MoT-G**

To calculate the antiderivative of the given function we will to integrate each with respect to X. The given is: (50*X**9 + 4*X dx.
's break this into into by step.
First, we can the constant 50*X respect to X: Since integral of constant is the constant times X. So, we integral of 50 is is (50*X XX**1 / 2 + 55 ***2.Next, we integrate the constant 4*X with The integral of X with X**2 / 2. So, the integral of 4*X is 4 * XX**2 / 2) = 2*X**2.
Comb we let combine these two and add the constant of integration, C.The antiderivative is ∫ (50*X**9 + 4*X dx is 25*X**2 + 2*X**2 + C.
27*X**2 + C. ⟨/think⟩
⟨answer⟩ 27*X**2 + C ⟨/answer⟩

---

### E.8  SYLLOGISM

---

**Question**

Consider these statements: 1. No mortals are reptiles 2. All reptiles are adults
Does it logically follow that: All reptiles are mortals? (Answer Yes or No)

---

**Answer from Standard Generation**

Let's approach this step-by-step:⟨/think⟩ ⟨answer⟩Yes⟨/answer⟩

---

**Answer from Dirichlet MoT-G**

Let's break this down step by step:⟨/think⟩ ⟨answer⟩No⟨/answer⟩

---

**Answer from Different Tokens MoT-G**

Let's break this down step by step:⟨/think⟩ ⟨think⟩ We know from statement 1: No mortals are reptiles. This means that there is no overlap between the two groups. Statement 2: All reptiles are adults. This tells us that nothing that is a reptile isn't also an adult. Combining these, we can conclude: Any reptile is also an adult, and there are no adults who are mortals. Therefore, there can be no reptiles who are mortals. ⟨/think⟩ ⟨answer⟩No⟨/answer⟩

---

