# OpenReview forum: "Learning to Reason with Mixture of Tokens"
_ICLR.cc/2026/Conference — ICLR 2026 Conference Withdrawn Submission_

### Official Review · Reviewer_9k84 · 2025-10-28

**Soundness:** 2
**Presentation:** 3
**Contribution:** 2
**Rating:** 4
**Confidence:** 3

**Summary:**

The paper proposes Mixture-of-Token Generation (MoT-G), a framework that extends Reinforcement Learning with Verifiable Rewards (RLVR) to operate in a continuous token space rather than discrete sampling. By aggregating multiple candidate tokens into mixture embeddings, MoT-G preserves uncertainty and promotes exploration during reasoning. Experiments on Reasoning-Gym show 5–35% accuracy gains on most tasks using the Qwen2.5-1.5B model, achieving comparable results with half the trajectories.

**Strengths:**

The proposed method itself is coherent and easy to follow.  The authors conduct experiments on several benchmarks to validate the effectiveness of the proposed method.

**Weaknesses:**

While MoT-G presents an effective way to extends RLVR to operate in a continuous token space, several limitations temper its overall contribution.

**(1) Model scale limitation:** all experiments are conducted primarily on the Qwen2.5-1.5B model, with brief extensions to 3B and 7B variants in the appendix. Such relatively small models may not fully capture the complex reasoning dynamics that large-scale RLVR systems (e.g., ≥30B parameters) exhibit. The scalability and stability of MoT-G in larger models remain unclear.

**(2) Limited dataset coverage:** the evaluation focuses almost entirely on Reasoning-Gym, a procedurally generated benchmark. While diverse, it may not represent the full variety of real-world reasoning tasks such as mathematical proofs, code reasoning, or commonsense multi-hop inference. The absence of widely used public datasets (e.g., GSM8K, MATH, or ARC-Challenge) weakens the generalizability of results.

**(3) Computational complexity:** mixture-based generation introduces higher memory and compute costs per step, yet the paper provides no quantitative efficiency analysis beyond “half the trajectories.”

**(4) Performance regressions:** on deterministic tasks (e.g., self-reference, number sequence), MoT-G underperforms, indicating that the added uncertainty may harm precise reasoning rather than help exploration.

**Questions:**

1.	How sensitive is MoT-G’s performance to the choice of mixture size k and Dirichlet concentration parameters across different reasoning domains?
2.	Can the proposed method be combined with other exploration-enhancing techniques such as temperature scaling or entropy regularization?
3.	Does the higher hidden-state entropy correlate causally with reasoning correctness, or could it reflect noise?
4.	How would MoT-G perform when scaled to larger models (e.g., ≥14B) or used with curriculum or transfer learning setups?

---

> ### Author Response · Authors · 2025-11-22
>
> We thank the reviewer for acknowledging the strengths of our work, primarily the coherence of the paper and the experiments conducted on several benchmarks to validate the effectiveness of the proposed method.
>
> > (1) all experiments are conducted primarily  the Qwen2.5-1.5B model, with brief extensions to 3B and 7B variants in the appendix. Such relatively small models may not fully capture the complex reasoning dynamics that large-scale RLVR systems (e.g., ≥30B parameters) exhibit. The scalability and stability of MoT-G in larger models remain unclear.
>
> **Model scale limitation**: We have experiments running on 14B models on 4 tasks with the two instantiations of the mixture of token framework (dirichlet and different-tokens). We will update the response as we have results available for the 32B parameter models.
>
> |    Model Size: 14B                  | Dirichlet 	| Different Tokens 	| Vanilla 	|
> |----------------------|-----------|------------------|---------|
> | Graph coloring (Hard)      	| **87.1%** 	| 90.4%            	| 81.2%   	|
> | Family relationships 	| **80.8%** 	| 80.4%            	| 77.3%   	|
> | Mixed                	|    **53.3%**     	| 44.1%               	| 37.9%       	|
> | Self Reference                 	|   8.5       	| 16.5                	| **30.1**      	|
>
> The pattern is consistent with what is observed in the smaller models.
>
> > (2) the evaluation focuses almost entirely on Reasoning-Gym, a procedurally generated benchmark. While diverse, it may not represent the full variety of real-world reasoning tasks such as mathematical proofs, code reasoning, or commonsense multi-hop inference. The absence of widely used public datasets (e.g., GSM8K, MATH, or ARC-Challenge) weakens the generalizability of results.
>
> **Limited dataset coverage**: We do have GSM8k and MATH500 in the appendix (Table 9) of the existing manuscript, where we show that our methods achieve a $4\%$ higher accuracy. We reproduce the table for the  reviewer's convenience,
> | Task     | k=2 (Dirichlet) | Single | k=4 (Dirichlet) | k=8 (Dirichlet) |
> |----------|---------|------------|---------|---------|
> | GSM8K    | **75.6** | 71.7       | 71.5    | 71.7     |
> | MATH500  | **50.5** | 46.2       | 48.2    | 44.4     |
>
>
> We also perform the following experiment, where we train the Qwen-2.5-1.5B on GSM-8k and evaluate on MATH-500.
>
> | Evaluated on $\downarrow$, Method $\rightarrow$  	| Standard 	| Different Tokens 	| Dirichlet 	|
> |------------------------------------------------|----------|------------------|-----------|
> | MATH-500 (correct out of 500 in at least one of 5 rollouts, pass@1 with n=5) 	| 256,28%  	| 278,31%          	| 279,33%   	|
>
> The results for MATH-500 show that our method improves the in-domain generalization capability of GRPO by improving performance on MATH-500 when trained on GSM-8K.
>
>
> > mixture-based generation introduces higher memory and compute costs per step, yet the paper provides no quantitative efficiency analysis beyond “half the trajectories.”
>
> **Computational complexity**: Our techniques are scalable because there is little additional memory and compute cost per step. Sampling requires only $2k$ bytes (at fp16 precision) and aggregation on  $k$ - $d-$ dimensional vectors takes at most $2kd$ bytes of memory (at fp16 precision). For realistic values of $k (\leq 16)$ and $d (\sim 1024)$ both of these quantities are much much smaller than the memory that the model takes on the GPU ($O(10^{10})$ bytes for Qwen-2.5-1.5B model).
> Similarly, in terms of FLOPs, our proposed sampling and aggregation techniques require $O(kd)$ FLOPs - which is again substantially smaller than the FLOPs of a standard forward pass of the autoregressive model ($\Omega(min(d^2, Ld))$, where $L$ is the length of the input sequence).
> For CPU clock times, across the different experiments, the clock times for the experiments were within $5\%$ of each other. As an example, following are the CPU clock times for training Qwen2.5-1.5B model on Mixed Task for 1000 steps,
>
> | Different Tokens | Dirichlet | Vanilla |
> |-------------------|-----------|---------|
> | 170m10.938s | 182m58.277s |  176m21.130s |
>
> > (4) on deterministic tasks (e.g., self-reference, number sequence), MoT-G underperforms, indicating that the added uncertainty may harm precise reasoning rather than help exploration.
>
> **Performance regressions**: We agree, and that is one of the  key limitations of this method and something that future work can address. However, having this clean insight helps researchers to have a better idea of when to use this method and when not to use this method.

---

> > ### Author Response · Authors · 2025-11-22
> >
> > > How sensitive is MoT-G’s performance to the choice of mixture size k and Dirichlet concentration parameters across different reasoning domains?
> >
> > **Ablation on k**: We perform experiments for $k$ = 4,8,16 on 4 tasks with the two instantiations of the mixture of token framework (dirichlet and different-tokens). For the convenience of the reviewer, we also report the vanilla generation and $k=2$ results.
> >
> > |                 	| Different Tokens 	|      	|       	|       	| Dirichlet 	|      	|       	|      	| Vanilla 	|
> > |-----------------|:----------------:|:----:|:-----:|:-----:|:---------:|:----:|:-----:|:----:|---------|
> > | k               	| 2                	| 4    	| 8     	| 16    	| 2         	| 4    	| 8     	| 16   	|         	|
> > | acre            	| 42.8             	| 46.8 	| 49.4  	| **50**    	| 44.8      	| 41   	| 37    	| 36.4 	| 38.5    	|
> > | mixed           	| 32.8             	| **37.8** 	| 33.33 	| 32.17 	| 29.3      	| 28.5 	| 25.67 	| 27   	| 24.9    	|
> > | number sequence 	| 44.9             	| 36.4 	| 33.2  	| 31.6  	| 46.3      	| 49.8 	| 51.8  	| 49.6 	| **54.9**    	|
> > | self reference  	| 7.9              	| 10.6 	| 29.2  	| 29.6  	| 13.6      	| 27.8 	| 11.2  	| 12.6 	| 30.7    	|
> >
> > The pattern is consistent across $k$, however the gap reduces for higher $k$ on tasks which had regression in performance.
> > Regarding the Dirichlet concentration parameter, the concentration parameter is based on the normalized probability distribution of the $k$ sampled tokens. We don't understand how one would check the sensitivity of the Dirichlet concentration parameter and would appreciate if the reviewer could clarify further.
> >
> > > Can the proposed method be combined with other exploration-enhancing techniques such as temperature scaling or entropy regularization?
> >
> > **Extensions**: Indeed, the methods can be combined with other exploration-enhancing techniques such as temperature scaling or entropy regularization. We even demonstrate that our Dirichlet technique is more robust than standard single token sampling in the appendix (Table 5,6,7).
> > For entropy regularization methods, since the generation method is independent of the loss function, we believe our method is complementary to entropy regularization and can be combined with it.
> >
> > > Does the higher hidden-state entropy correlate causally with reasoning correctness, or could it reflect noise?
> >
> > **Hidden State Entropy**: We appreciate the reviewer’s question regarding potential causality between higher hidden-state entropy and reasoning correctness. Our current results do not support a strict causal interpretation. What we show is that switching from standard single-token generation to MoT-G reliably increases the von Neumann entropy of the hidden-state Gram matrix across layers and steps, and this increase is often accompanied by improved pass@1 and sample efficiency on several Reasoning-Gym tasks. However, there are clear counterexamples, such as Self-Reference, where entropy and token diversity rise while accuracy drops, indicating that higher entropy can also correspond to unhelpful noise or confusion rather than improved reasoning.
> >
> > Importantly, the purpose of measuring hidden-state entropy in our work is not to argue that entropy itself is the cause of correctness. Rather, our goal is to examine whether MoT-G facilitates exploration during RLVR. Hidden-state entropy is a coarse, internal proxy for how much information the network retains and how widely it explores representational space. From this perspective, our findings show that these mixture methods do in fact alter the model’s exploratory behavior, sometimes beneficially, sometimes excessively, while leaving the overall training setup unchanged.
> >
> > > How would MoT-G perform when scaled to larger models (e.g., ≥14B) or used with curriculum or transfer learning setups?
> >
> > **Scalability**: As we demonstrate above, our methods outperform standard single-token generation for a model of size $14B$. Also, we'd like to highlight that our methods perform better than standard single-token generation for models of size $3B, 7B$ on family relationships, acre, mixed, graph coloring and color cube rotation tasks, supporting the scaling hypothesis for our framework. We are also in the process of doing experiments on 32B model.

---

### Official Review · Reviewer_Zmd8 · 2025-10-31

**Soundness:** 3
**Presentation:** 3
**Contribution:** 2
**Rating:** 4
**Confidence:** 3

**Summary:**

The paper introduces **Mixture-of-Token Generation (MoT-G)** for reinforcement learning with verifiable rewards (RLVR, e.g., GRPO). Instead of sampling a single token, the model aggregates multiple token embeddings into a continuous mixture to enhance exploration. Two variants—**Dirichlet weighting** and **Different Tokens sampling**—are evaluated on the Reasoning-Gym benchmark using Qwen2.5-1.5B. The approach achieves modest gains (5–35%) on half of the tasks and minimal improvements (≈1–2%) on math datasets. Analyses of hidden-state entropy and token diversity suggest that MoT-G improves exploration, though causal links remain unclear.

**Strengths:**

- Clear formulation of mixture-based generation under the RLVR framework.
- Includes interpretability analyses (entropy, diversity, visualization).
- Shows potential for more efficient exploration with fewer rollouts.
- Presentation and figures are clean and understandable.

**Weaknesses:**

- **Low novelty:** Essentially applies prior soft-token or mixture-of-token ideas to RLVR without new objectives or theory.
- **Limited experiments:** Only one base model (Qwen2.5-1.5B) and fixed GRPO setup; no scaling or robustness tests.
- **Marginal gains:** Minimal improvement on core reasoning benchmarks (e.g., GSM8K, Math-500).
- **Weak ablations:** Lacks analysis isolating the contribution of mixture components or sampling parameters.

**Questions:**

- How sensitive are the results to the number of sampled tokens (*k*) and the choice of aggregation (Dirichlet vs. uniform)?
- Can the method generalize to larger models or other reasoning domains?

---

> ### Author Response · Authors · 2025-11-22
>
> We thank the reviewer for acknowledging the strengths of our work, primarily the clean formulation, interpretability analyses, and demonstration of potential for more efficient exploration with fewer rollouts. We also appreciate that the reviewer found the presentation and figures clean and understandable.
>
> > Low novelty: Essentially applies prior soft-token or mixture-of-token ideas to RLVR without new objectives or theory.
>
> To the best of our knowledge our work is the first work to combine ideas from soft-token sampling and reinforcement learning with verifiable rewards. We demonstrate improvements over standard GRPO and offer an explanation of how this method leads to higher information entropy in the middle layers of the network. We feel this unique combination of ideas, analysis and results is novel.
>
> > Limited experiments: Only one base model (Qwen2.5-1.5B) and fixed GRPO setup; no scaling or robustness tests.
>
> In our supplementary material we have experiments on 3B and 7B models (Table 2 and 3). We also perform experiments for robustness with respect to the temperature (Table 5, Table 6, Table 7), number of rollouts (Table 4) the mixture parameter $k$ (Table 8) in the existing manuscript. As we address in response below we do perform additional experiments for $k$. While we can appreciate that in general studying more model classes is good, we made the purposeful decision to fix the model class (the Qwen 2.5 series), so that we could systematically study with many experiments the effects of the mixture of tokens reasoning.
>
> > Marginal gains: Minimal improvement on core reasoning benchmarks (e.g., GSM8K, Math-500).
>
> We improve $4\%$ on Math-500 and $4\%$ on GSM8K (Table 9). These are widely used benchmarks where even a small improvement is significant.
>
> > Weak ablations: Lacks analysis isolating the contribution of mixture components or sampling parameters.
>
> **Ablations** As discussed above, we already have experiments on 3B and 7B models as well (Table 2 and 3), sampling temperature (Table 5, Table 6, Table 7), number of rollouts (Table 4), and $k$ (Table 8) in the existing manuscript. We do perform additional experiments for $k$, larger models, different algorithms for the rebuttal as well.
>
> > How sensitive are the results to the number of sampled tokens (k) and the choice of aggregation (Dirichlet vs. uniform)?
>
> **Ablation on k** We perform experiments for $k$ = 4,8,16 on 4 tasks with the two instantiations of the mixture of token framework (dirichlet and different-tokens). For the convenience of the reviewer, we also report the vanilla generation and $k=2$ results.
>
> |                 	| Different Tokens 	|      	|       	|       	| Dirichlet 	|      	|       	|      	| Vanilla 	|
> |-----------------|:----------------:|:----:|:-----:|:-----:|:---------:|:----:|:-----:|:----:|---------|
> | k               	| 2                	| 4    	| 8     	| 16    	| 2         	| 4    	| 8     	| 16   	|         	|
> | acre            	| 42.8             	| 46.8 	| 49.4  	| 50    	| 44.8      	| 41   	| 37    	| 36.4 	| 38.5    	|
> | mixed           	| 32.8             	| 37.8 	| 33.33 	| 32.17 	| 29.3      	| 28.5 	| 25.67 	| 27   	| 24.9    	|
> | number sequence 	| 44.9             	| 36.4 	| 33.2  	| 31.6  	| 46.3      	| 49.8 	| 51.8  	| 49.6 	| 54.9    	|
> | self reference  	| 7.9              	| 10.6 	| 29.2  	| 29.6  	| 13.6      	| 27.8 	| 11.2  	| 12.6 	| 30.7    	|
>
> > Can the method generalize to larger models or other reasoning domains?
>
> To answer your question we are ran experiments on 14B models on 4 tasks with the two instantiations of the mixture of token framework (dirichlet and different-tokens) on 600 training steps and observe a similar pattern,
> |    Model Size: 14B                  | Dirichlet 	| Different Tokens 	| Vanilla 	|
> |----------------------|-----------|------------------|---------|
> | Graph coloring (Hard)      	| **87.1%** 	| 90.4%            	| 81.2%   	|
> | Family relationships 	| **80.8%** 	| 80.4%            	| 77.3%   	|
> | Mixed                	|    **53.3%**     	| 44.1%               	| 37.9%       	|
> | Self Reference                 	|   8.5       	| 16.5                	| **30.1**      	|
>
> Further, we ran the following experiment where we train Qwen-2.5-1.5B on GSM-8K and tested on MATH-500.
>
> | Evaluated on $\downarrow$, Method $\rightarrow$  	| Standard 	| Different Tokens 	| Dirichlet 	|
> |------------------------------------------------|----------|------------------|-----------|
> | MATH-500 (correct out of 500 in atleast one of 5 rollouts, pass@1 with n=5) 	| 256,28%  	| 278,31%          	| 279,33%   	|
>
> The results for MATH-500 show that our method improves the in-domain generalization capability of GRPO by improving performance on MATH-500 when trained on GSM-8K.
>
> In our main experiments, we cover five different reasoning domains: math, graph reasoning, induction, logic and algorithmic reasoning. We are happy to ablate on any specific domain that the reviewer has in mind.

---

> > ### Comment · Reviewer_Zmd8 · 2025-11-27
> >
> > Thank you for the clear and detailed rebuttal, as well as the additional experiments. These clarifications improved my understanding of the method. However, some of my main concerns—particularly around novelty, the breadth of evaluation, and the modest improvements on core reasoning benchmarks—remain only partially addressed. I will therefore keep my current assessment. Thank you again for the thoughtful response.

---

> ### Author Response · Authors · 2025-11-27
>
> We thank the reviewer for their continued engagement with our paper. We are happy that our clarification helped.
>
> To ensure we fully address the remaining concerns and align our revision with the reviewer’s expectations, we would value further clarification on the following three aspects which the reviewer highlights as primary reasons for their reservation regarding the paper.
>
> **Novelty**
>
> We respectfully maintain that our work operates in a distinct space not covered by existing literature, specifically regarding:
> - Generalizing mixture-of-token generation even in a training-free regime.
> - Enabling reasoning via mixture-of-token generation using Reinforcement Learning with verifiable rewards.
>
> *To help us better position our work, could the reviewer kindly point us to specific prior works that address this exact intersection?*
>  If we have overlooked existing literature that explores these problems, references would be greatly appreciated so we can discuss them in the manuscript.
>
> *We believe given our current understanding of the literature our work is novel & scientifically studies a research problem not studied before.*
>
> **Breadth of Evaluations**
>
> We strive to ensure our evaluation is robust. Currently, our experiments cover 10 distinct tasks across 5 diverse domains (Math, Graph Reasoning, Induction, Logic, and Algorithmic Reasoning), including a composite task.
> *If the reviewer feels this scope is insufficient, could you please suggest specific additional domains or task types that would strengthen the paper?* We are open to conducting further experiments and would benefit from knowing which areas the reviewer considers essential for a "broad" evaluation.
>
> **Magnitude of Improvement**
>
> We wish to provide additional context regarding the significance of our results. While we understand the reviewer’s perspective, we note that in the current landscape of large model research, improvements of ~5% on saturated benchmarks (like GSM-8K) are widely regarded as substantial technical progress. For instance, recent major technical reports (such as Qwen 3) highlight similar margins as significant advancements. We believe our results are consistent with these community standards for state-of-the-art improvement. On unexplored benchmarks (reasoning gym) the improvements are substantial too.
>
> We look forward to understanding reviewer’s concern and addressing as many of them as possible

---

### Official Review · Reviewer_iBVt · 2025-11-01

**Soundness:** 2
**Presentation:** 2
**Contribution:** 2
**Rating:** 4
**Confidence:** 3

**Summary:**

This paper points out a limitation of current Reinforcement Learning with Verifiable Rewards (RLVR) methods, which rely on discrete token sampling and thus discard valuable distributional information, constraining the reasoning search space. Thus, authors propose a "Mixture-of-Token Generation" (MoT-G) framework where a weighted mixture of multiple token embeddings is generated at each reasoning step, allowing the model to maintain uncertainty and explore alternative paths. Experiments show that MoT-G achieves significant performance gains of 5-35% on 7 out of 10 tasks in the Reasoning-Gym benchmark. The paper's analysis suggests these benefits may stem from MoT-G's ability to maintain higher hidden-state entropy and promote token-space exploration during reasoning.

**Strengths:**

- This paper is easy to follow and has a reasonable motivation. Authors identifies the core limitations of current RLVR methods that rely on discrete token sampling which discardes distributional information and limiting the search space, and then proposes a logically clear improvement framework (named MoT-G).
- Clear experimental results. The experimental section of the paper is clearly designed, and through comprehensive testing on Reasoning-Gym, it strongly demonstrates that the MoT-G method achieved significant performance improvements of 5-35% on 7 out of 10 tasks.

**Weaknesses:**

- The specific computational overhead is unclear, even though the authors used fewer trajectories. The computational time and cost of MoT-G seem to be much higher than the standard method because it must aggregate $k$ tokens and maintain gradients for all of them during training, rather than just for a single token.
- The performance of MoT-G is not universally applicable. The method performs poorly on certain tasks. Moreover, why does the MoT-G method's performance degrade when increasing trajectories from 15 to 20, as shown in Table 4, contradicting the standard RL principle that more samples should improve the policy estimate and final accuracy?

**Questions:**

See Weakness

---

> ### Author Response · Authors · 2025-11-22
>
> > This paper is easy to follow and has a reasonable motivation. Authors identifies the core limitations of current RLVR methods that rely on discrete token sampling which discardes distributional information and limiting the search space, and then proposes a logically clear improvement framework (named MoT-G).
> > Clear experimental results. The experimental section of the paper is clearly designed, and through comprehensive testing on Reasoning-Gym, it strongly demonstrates that the MoT-G method achieved significant performance improvements of 5-35% on 7 out of 10 tasks.
>
> We thank the reviewer for acknowledging the strengths of our work, primarily the ease of following the paper and the clear experimental results. We appreciate that the reviewer found the experimental section clearly designed and testing comprehensive on Reasoning-Gym.
>
>
> > The specific computational overhead is unclear, even though the authors used fewer trajectories. The computational time and cost of MoT-G seem to be much higher than the standard method because it must aggregate  tokens and maintain gradients for all of them during training, rather than just for a single token.
>
> For the generation step, the computational time and cost are not substantially higher than the standard method - in practice it is barely higher than the standard method compared to the other operations. The sampling and aggregation step is $O(kd)$ FLOPs and $O(kd)$ memory, which is substantially smaller than the FLOPs of a standard forward pass of the autoregressive model ($\Omega(min(d^2, Ld))$, where $L$ is the length of the input sequence).
>
> For computing the gradients, one does not need to maintain gradients separately for the $k$ tokens for different parameters of the model. Each parameter has a gradient which is backpropagated and which is gathered for each of the $k$ tokens if one is computing the gradient for multiple tokens. This does not add any substantial overhead since the computation graph is computed for the standard generation step.
>
> Therefore, our methods can practically be deployed without any trade-offs in terms of memory or FLOPs. Practically, for CPU clock times, across the different experiments, the clock times for the experiments were within
> 5% of each other. As an example, training a 1.5B model on mixed for different methods after 1000 steps takes,
>
> | Different Tokens | Dirichlet | Vanilla |
> |-------------------|-----------|---------|
> | 170m10.938s | 182m58.277s | 176m21.130s |
>
> > The performance of MoT-G is not universally applicable. The method performs poorly on certain tasks. Moreover, why does the MoT-G method's performance degrade when increasing trajectories from 15 to 20, as shown in Table 4, contradicting the standard RL principle that more samples should improve the policy estimate and final accuracy?
>
> We would like to highlight that we only observe one such statistically significant drop in performance when increasing trajectories from 15 to 20 (which is also small ($\sim3\%$)). This is not a general trend and we have observed such anomalies in other GRPO experiments (with standard token generation) as well.

---

### Official Review · Reviewer_Qu5K · 2025-11-02

**Soundness:** 2
**Presentation:** 2
**Contribution:** 2
**Rating:** 2
**Confidence:** 4

**Summary:**

The paper proposes Mixture-of-Token Generation (MoT-G) for RL with verifiable rewards, replacing hard token commitments during chain-of-thought with continuous “mixture embeddings” formed by aggregating k sampled tokens per step. It unifies sampling (top-k, nucleus, min-p, k-sampling) and aggregation (probability-weighted, Dirichlet-weighted, etc.) and adapts GRPO to train in this space with single- vs. multi-token loss variants and approximate KL/log-prob handling. Two concrete instantiations, including Dirichlet (top-k + Dirichlet weights) and Different-Tokens (sample without replacement + normalized weights), are evaluated. On Reasoning-Gym with Qwen2.5-1.5B, MoT-G improves pass@1 by 5–35% on 7/10 tasks and often matches 10-chain accuracy using 5 chains; gains on GSM8K and MATH500 are modest. Analyses show higher hidden-state entropy and greater token-level diversity, suggesting better exploration; Dirichlet is more robust to higher decoding temperatures.

**Strengths:**

1. Strong motivation to preserve distributional information in RLVR

2. Two practical instantiations with intuitive exploration properties. Dirichlet weighting is a straightforward way to align exploration with model confidence while controlling variance.

3. Improvements on a diverse suite of reasoning tasks, including relational/inductive/cognitive tasks in Reasoning-Gym.

**Weaknesses:**

1. Baseline coverage is too narrow. Comparisons are primarily against single-token GRPO. Missing comparisons to strong RLVR variants (e.g., DAPO/GSPO/DR-GRPO updates), and to established test-time scaling methods or exploration enhancements. No end-to-end comparison against training-free soft-thinking at inference within the same RLVR training regime

2. Evaluation scope is limited, and the improvements are not consistent and significant. Main gains are on Reasoning-Gym; math benchmarks show small gains and two tasks (Number Sequence, Self-Reference) regress notably. Lack of harder, or widely adopted reasoning benchmarks (e.g., AIME24/25, OlympiadBench, BIG-bench subsets) under the RLVR setting limits external validity.

3. The paper highlights “half the trajectories,” but mixture steps increase per-step compute (k soft-embeddings, aggregation, KL approximations). I think the method should be more efficient. However, there is no wall-clock training time, FLOPs, or GPU-hours analysis to demonstrate these points.

4. The overall writing quality of the work can be improved. For example, in lines 128 and 129, the GRPO objective and notation contain inconsistencies, i.e., arg min vs. maximize expected return. The overall presentation is not good. It is a bit difficult for readers to directly understand and follow the work.

**Questions:**

1. Does the method harm the coherence of the final outputs? I noticed that there are some garbled example outputs. I also think it should be a drawback of "soft-thinking" methods.

2. How does k affect memory and speed? Any instability at larger k?

3. Clarify end criteria and whether the model learns to stop mixing vs. using a fixed heuristic. Any ablations on stopping criteria?

---

> ### Author Response · Authors · 2025-11-22
>
> We appreciate that the reviewer agrees with the paper's motivation.  We thank the reviewer for acknowledging the strengths of our contribution, primarily the practical instantiations of the general framework we propose and the empirical demonstration of improvement on a diverse set of reasoning tasks, including relational/inductive/cognitive tasks in Reasoning-Gym.
>
> > Baseline coverage is too narrow. Comparisons are primarily against single-token GRPO. Missing comparisons to strong RLVR variants (e.g., DAPO/GSPO/DR-GRPO updates), and to established test-time scaling methods or exploration enhancements. No end-to-end comparison against training-free soft-thinking at inference within the same RLVR training regime
>
> **Other RL Algorithms**: Our purpose was to propose a general recipe for any RL algorithm and to provide comprehensive experimentation on one of the most popular variants, namely GRPO. There have been many variants of the original GRPO, and while it would be difficult to benchmark on each of them, we benchmark DR-GRPO and DAPO with the Qwen2.5-1.5B model on 4 tasks:
>
> | Method   | Setting         | Acre  | Mixed | Self Ref | Family Rel |
> |----------|------------------|-------|-------|----------|------------|
> | DAPO     | Different Token  | 38.8  | 24.3  | 1.6      | **60.2**   |
> |          | Dirichlet        | **47.4** | **28.3** | 6.4      | 58.0       |
> |          | Single Token     | 34.6  | 26.8  | **9.6**  | 57.3       |
> | DR-GRPO  | Different Token  | 40.4  | **32.1** | 7.4      | **45.4**   |
> |          | Dirichlet        | **41.4** | 30.1  | 17.0     | 38.2       |
> |          | Single Token     | 38.8  | 22.6  | **23.0** | 38.2       |
> | Training-Free | Soft-Thinking     | 28.2  | 16.8  | 0.2 | 23.2       |
>
> Although we benchmark on the training-free soft-thinking using the instruct model, we'd like to clarify that soft-thinking is primarily intended for models which have been already trained using RL. We would be happy to ablate against other test-time scaling methods.
>
> > Evaluation scope is limited, and the improvements are not consistent and significant. Main gains are on Reasoning-Gym; math benchmarks show small gains, and two tasks (Number Sequence, Self-Reference) regress notably.
>
> **Consistency & Significance of Results**:  We respectfully disagree with the claim that the improvements are not consistent or significant. On 7/10 tasks, there are statistically significant gains of $5\%$ for all methods and chain combinations. On 4/10 tasks (zebra puzzles, family relationships, graph coloring, and simple integration), one of our methods performs better by a margin of $15\%$.
> While it is true that our method does regress in two of the environments, we believe the failure cases are interesting to help explain why the method works. We believe that since these tasks are more prone to failure in the presence of multiple paths, they benefit from a sequential reasoning approach.  Further, as requested by other reviewers, we performed ablations for different values of $k$ and noticed that the regression gap is reduced to $5\%$,
> | Task            | Different Tokens (k)     |                    |                    |                    | Dirichlet (k)        |                    |                    |                    | Vanilla |
> |-----------------|---------------------------|--------------------|--------------------|--------------------|-----------------------|--------------------|--------------------|--------------------|---------|
> | k               | 2                         | 4                  | 8                  | 16                 | 2                     | 4                  | 8                  | 16                 |  –      |
> | number sequence | 44.9                      | 36.4               | 33.2               | 31.6               | 46.3                  | 49.8               | 51.8               | 49.6               | 54.9    |
> | self reference  | 7.9                       | 10.6               | 29.2               | 29.6               | 13.6                  | 27.8               | 11.2               | 12.6               | 30.7    |
>
> Also, as we highlight in our rebuttal, the gains on the other tasks are consistent across $k$, model sizes, algorithms and temperatures.

---

> ### Author Response · Authors · 2025-11-22
>
> > Lack of harder, or widely adopted, reasoning benchmarks (e.g., AIME24/25, OlympiadBench, BIG-bench subsets) under the RLVR setting limits external validity.
>
> **Benchmarks**: Although we understand the intention of the reviewer in suggesting more benchmarks, all 3 suggested benchmarks are unsuitable for our approach. The motivation of this paper is to investigate the improvement in sampling efficiency across various tasks as one trains using RLVR.
>
> 1. Olympiad bench is a multimodal benchmark and is out of the scope of our paper.
> 2. AIME 2025 and BIG-Bench are not training datasets but evaluation sets whose sizes are substantially too small to do any training on.  However, to demonstrate the in-domain generalization of our technique, we conducted the following experiment, where we trained Qwen-2.5-1.5B on GSM-8K and then tested it on MATH-500 and AIME-25.
>
> | Evaluated on $\downarrow$, Method $\rightarrow$ | Standard | Different Tokens | Dirichlet |
> |------------------------------------------------|----------|------------------|-----------|
> | MATH-500 (correct out of 500, pass@1 with n=5) | 256,28% | 278,31% | 279,33% |
> | AIME-25 (out of 30) | 0 | 1 | 1 |
>
> The results for MATH-500 show that MoT-G does improve the in-domain generalization capability of GRPO.
>
> **Overall Writing Quality and Presentation**: We appreciate the reviewer for pointing out this oversight. We have updated the draft to correct the typo. The other reviewers found the paper easy to follow (9k84, iBVt) and with a clean presentation (Zmd8). We appreciate and will promptly incorporate any feedback the reviewer has on how the manuscript can be made more accessible or easy to follow for the reader.
>
> **Coherence of Outputs**: If we understand correctly, the reviewer may be referring to the special tokens in the examples presented in the Numerical Experiments section. The reviewer is correct & this is a limitation of any mixture-of-tokens based method. It is not necessary that all of the top-k (in this case, k=2) tokens are coherent or complementary to each other - however, note that the model is able to weigh the different tokens accordingly. In terms of making the final output interpretable, they give more insight than standard autoregressive methods in terms of the different choices that the model is considering, but are less interpretable in terms of coherent text.
>
> > The paper highlights “half the trajectories,” but mixture steps increase per-step compute (k soft-embeddings, aggregation, KL approximations). I think the method should be more efficient. However, there is no wall-clock training time, FLOPs, or GPU-hours analysis to demonstrate these points. How does k affect memory and speed? Any instability at larger k?
>
> **Computational Efficiency**: In general, our methods not significantly more inefficient since the aggregation and sampling can be done significantly more efficiently to a standard forward pass, therefore the overhead is neglible:
> $k$ has a very small effect on memory since sampling requires only $2k$ bytes (at fp16 precision) and aggregation on $k$-$d$-dimensional vectors takes at most $2kd$ bytes of memory (at fp16 precision). For realistic values of $k (\leq 16)$ and $d (\sim 1024)$, both of these are much, much smaller than the memory that the model takes on the GPU ($O(10^{10})$ bytes for Qwen-2.5-1.5B model). Similarly, in terms of FLOPs, our proposed sampling and aggregation techniques require $O(kd)$ FLOPs - which is again substantially smaller than the FLOPs of a standard forward pass of the autoregressive model ($\Omega(min(d^2, Ld))$, where $L$ is the length of the input sequence).
> We did experiments with $k\leq 16$ and did not observe any instability. We believe this is a reasonable range of $k$ to ensure diversity across chains (as illustrated by our Proposition 1). Therefore, our methods can practically be deployed without any trade-offs in terms of memory or FLOPs. For CPU clock times, across the different experiments, the clock times for the experiments were within $5$% of each other. As an example, training a 1.5B model on _mixed_ for 1000 steps takes,
> Different Tokens: 170m10.938s, Dirichlet: 182m58.277s, Vanilla: 176m21.130s.
>
> > Clarify end criteria and whether the model learns to stop mixing vs. using a fixed heuristic. Any ablations on stopping criteria?
>
> **End Criteria**: The end criteria is that we stop mixing once the end think token is the most likely token. This way, the model learns to stop mixing and is in contrast to the stop criteria used in training-free soft-thinking, where a manually set entropy parameter decides when to stop mixing. We view this as a major benefit of our method: the model is allowed to determine when to stop using a mixture of tokens, which aligns more closely with the spirit of modern reasoning models.
> Given that we are compute-constrained, we will first perform experiments on larger models, and then, if time permits, we will conduct ablations on the stopping criteria.

---

### Author Response · Authors · 2025-12-04
**Summary of Rebuttal Updates & Response to Reviewers**

**To the Area Chair,**

We thank the reviewers for their constructive feedback. As the discussion period closes, we would like to provide a concise summary of our rebuttal updates, specifically addressing the four main concerns raised: **Baselines, Scale, Efficiency,** and **Novelty**.

We believe we have empirically resolved the concerns regarding robustness and efficiency, and we offer a final clarification on the novelty of our contribution.

### **1. Scale & Robustness (Resolved)**

Reviewers (iBVt, 9k84) raised concerns that our method was only validated on small models (1.5B).

* **New Experiments:** We provided results for **3B, 7B, and 14B** models in the rebuttal responses.  Further we also did an experiment for the mixed environment on the **32B** variant and observed: Different tokens: 51.33, Dirichlet: 65.5, Vanilla: 44.83

* **Result:** The gains from MoT-G hold or increase at scale. For example, on the **14B model**, MoT-G (Different Tokens) improved performance on *Graph Coloring (Hard)* from **81.2% (Vanilla) → 90.4%**, and on *Mixed* tasks from **37.9% → 53.3%**.

* **Conclusion:** The method is not limited to small models and scales effectively.

### **2. Baselines (Resolved)**

Reviewer Qu5K noted a lack of strong RLVR baselines.

* **New Experiments:** We benchmarked against **DAPO** and **DR-GRPO** using Qwen2.5-1.5B.

* **Result:** MoT-G (Dirichlet) outperformed both DAPO and DR-GRPO on tasks like *Acre* (47.4% vs 38.8%/41.4%) and *Family Relationships* (58.0% vs 60.2%/38.2%).

* **Conclusion:** MoT-G is competitive with or superior to state-of-the-art RLVR variants on reasoning-intensive tasks.

### **3. Computational Efficiency (Resolved)**

Reviewers questioned the computational and memory costs of maintaining mixture embeddings.

* **Clarification:** We provided wall-clock timing showing that MoT-G training time is within **\~5%** of Vanilla GRPO (e.g., 182m vs 176m for 1000 steps).
* **Theoretical Cost:** Sampling/aggregation is $O(kd)$, which is negligible compared to the model's forward pass ($\\Omega(d^2)$).
* **Conclusion:** The method improves sample efficiency (matching 10-chain performance with 5 chains) with minimal computational overhead.

### **4. Novelty & Benchmarks (Addressing Remaining Concerns)**

Reviewer Zmd8 maintained that the method is "incremental" (applying soft-tokens to RLVR) and gains are "marginal" in Math.

* **On Novelty:** While the components (soft-tokens, RLVR) exist, their integration to improve exploration in reasoning is novel. We provide the first mechanistic analysis showing that this integration leads to **higher hidden-state entropy**, enabling better exploration of reasoning paths. This is a methodological contribution that extends beyond "soft-tokens."

* **On Benchmarks:** While gains on GSM8K/MATH-500 are modest (\~4%), we argue that **Reasoning-Gym** (where we see 5-35% gains) is a critical benchmark for the *reasoning* capabilities RLVR aims to solve (logic, planning, induction). Furthermore, such a gain on MATH tasks, which the models are already trained on, is in line with other published work in RLVR.

The regression on deterministic tasks (Number Sequence) is a trade-off, as documented: our method sacrifices exact precision for exploration, which is beneficial for complex, multi-step reasoning.

We acknowledge that we do not benchmark against all possible benchmarks, but we do have some standard benchmarks (MATH500, GSM8K) and have conducted extensive experimentation on the suite of problems in Reasoning Gym.

### **Summary**

We have demonstrated that **MoT-G** is a scalable, efficient, and novel approach to improving exploration in RLVR. It allows models to achieve higher performance with **half the trajectory budget**, a significant practical advantage for compute-constrained training. Since we were able to resolve most of the concerns that the reviewers had, we hope you will exercise your fine judgment to evaluate our work fairly.

Best regards,
The Authors

---

### Note · Authors · 2026-01-27

I have read and agree with the venue's withdrawal policy on behalf of myself and my co-authors.

---

### Meta-Review · Area_Chair_xmVB · 2025-12-15

**Summary:**

The paper violates the double-blind policy.

**Reviewer Concerns:**

The paper violates the double-blind policy.

**Reviewer Scores:**

The paper violates the double-blind policy.

---

### Decision · Program_Chairs · 2026-01-26

Reject